# Cryoprotectant enables structural control of porous scaffolds for exploration of cellular mechano-responsiveness in 3D

Shumeng Jiang[1,4], Cheng Lyu[1,4], Peng Zhao[1], Wenjing Li[1], Wenyu Kong[1], Chenyu Huang [2], Guy M. Genin[3] & Yanan Du [1]

Despite the wide applications, systematic mechanobiological investigation of 3D porous scaffolds has yet to be performed due to the lack of methodologies for decoupling the complex interplay between structural and mechanical properties. Here, we discover the regulatory effect of cryoprotectants on ice crystal growth and use this property to realize separate control of the scaffold pore size and stiffness. Fibroblasts and macrophages are sensitive to both structural and mechanical properties of the gelatin scaffolds, particularly to pore sizes. Interestingly, macrophages within smaller and softer pores exhibit pro-inflammatory phenotype, whereas anti-inflammatory phenotype is induced by larger and stiffer pores. The structure-regulated cellular mechano-responsiveness is attributed to the physical confinement caused by pores or osmotic pressure. Finally, in vivo stimulation of endogenous fibroblasts and macrophages by implanted scaffolds produce mechano-responses similar to the corresponding cells in vitro, indicating that the physical properties of scaffolds can be leveraged to modulate tissue regeneration.

---

[1] Department of Biomedical Engineering, School of Medicine, Tsinghua-Peking Center for Life Sciences, MOE Key Laboratory of Bioorganic Phosphorus Chemistry and Chemical Biology, Tsinghua University, Beijing 100084, China. [2] Department of Dermatology, Beijing Tsinghua Changgung Hospital, School of Clinical Medicine, Tsinghua University, Beijing 102218, China. [3] Department of Mechanical engineering and Material Science, Washington University at St. Louis, St. Louis 63130, USA. [4]These authors contributed equally: Shumeng Jiang and Cheng Lyu. Correspondence and requests for materials should be addressed to Y.D. (email: duyanan@tsinghua.edu.cn)

Most adherent cells actively sense their surrounding extracellular matrix (ECM) and are concomitantly modulated by matrix properties in a reciprocal manner[1]. Recent advances in mechanobiology highlight the profound influence of ECM biophysical cues on a wide range of cell behaviors, such as cell growth, motility, differentiation, apoptosis, gene expression, adhesion, and signal transduction[2]. To date, most mechanobiological investigations on cell-biomaterial interactions have focused on 3D nano-porous hydrogels[3] or 2D substrates[4] with varied physical and biochemical properties such as stiffness, viscoelasticity[5], ligand density[6], and topography[7].

3D porous bioscaffolds with pore structures in the micrometer range are widely applied in tissue engineering to facilitate tissue regeneration (e.g., skin, bone, and cornea)[8–10]. Compared with hydrogels, the interconnected spaces in porous bioscaffolds provide a 3D microenvironment that facilitates a wider range of cellular activities, such as migration and cell–cell interactions. Despite their numerous potential therapeutic applications, systematic mechanobiological investigation on cellular behaviors in 3D porous bioscaffolds has yet to be performed owing to the lack of methodologies for decoupling the complex interplay between their structural and mechanical properties. For instance, the pore sizes of electrospun porous bioscaffolds can be manipulated by altering the fiber diameter and density between the interconnected fibers thereby influencing bulk stiffness[11]. For freeze–dried porous bioscaffolds, pore size and stiffness can be concomitantly controlled by tuning the freezing temperature[12] or altering the degree of chemical cross-linking for polymerization[13]. As alternative approaches to generate porous bioscaffolds, the particle-leaching or gas-forming methods can be used to produce pores with relatively precise sizes without altering their bulk mechanical properties; however, pore foaming approaches produce scaffolds with limited connectivity[14] and poor homogeneity owing to the undesired aggregation of porogens[15]. In addition, the bulk stiffness of the resulting scaffolds following porogen removal tends to be significantly weakened[16], limiting their application in situations that require sufficient stiffness to support the defective area. Together, these difficulties hinder the exploration of the mechano-responsiveness of cells in 3D porous bioscaffolds, which constitutes an obvious missing link in our understanding of cell–matrix interactions in mechanobiological studies. This missing link also highlights the unmet need for 3D porous scaffold fabrication strategies that enable the precise and separate control of specific mechanical and structural properties.

Here, a strategy for porous scaffold fabrication was explored with the aim of decoupling the structural and mechanical properties of the scaffold by cryoprotectant-enabled pore structural control. We accomplished systematic evaluation of cell mechano-responsiveness to independent changes in the mechanical or structural properties of scaffolds. Fibroblasts and macrophages were chosen for investigating mechano-responsiveness owing to their abundance and their important role in sensing mechanical stimuli and reconstituting the cellular microenvironment. Moreover, we explored in vivo cellular responses to these biophysically fine-tuned porous scaffolds after subcutaneous implantation, and our results highlight potential applications of the scaffolds in improving the therapeutic efficacy for regenerative medicine.

## Results

**Cryoprotectant-based structural control in cryogelated scaffold.** We discovered that the cryoprotectant DMSO can be used to control pore size through regulating ice crystal formation during fabrication of cryogelated 3D porous scaffolds (Fig. 1a). Thus, the pore sizes of the resulting 3D porous scaffolds were determined by the corresponding ice crystal sizes, whereas the stiffness of the scaffolds was regulated by the degree of cross-linking (e.g., by addition of the cross-linker glutaraldehyde (GA)) during cryogelation of the polymer. Regarding the regulatory mechanism of cryoprotectant-enabled pore size control, we hypothesized that DMSO enrichment during cryogelation alters the freezing point of the precursor solution and thus affects the corresponding ice crystal formation. Specifically, it is known that the concentration of the widely applied cryoprotectant DMSO in an aqueous solution determines the freezing point of the solution (Supplementary Fig. 1). During cryogelation, a precursor solution containing gelatin, GA and a low concentration of DMSO (e.g. 10%, with a freezing point > −10 ℃) was cooled to −20 ℃ to initiate ice crystal formation. In this scenario, ice crystal growth would lead to increased concentration of the remaining precursor solution (including DMSO) surrounding the ice crystals. Owing to DMSO enrichment, the freezing point of the remaining solution would drop and ultimately reach −20 ℃, at which point the DMSO concentration would rise to ~ 40% and further ice crystal growth would be halted. Therefore, by varying the initial DMSO concentration in the precursor solution, we should be able to control the duration of ice crystal growth until the DMSO reaches a concentration of ~ 40% when further freezing is halted, hence controlling the sizes of ice crystals and pores in the resulting cryogelated scaffolds (Fig. 1b). To verify our hypothesis, we first observed the enrichment of DMSO during ice crystal growth by applying hydrophobic oil red to specifically label DMSO distribution (Supplementary Video 1, Supplementary Fig. 2). During ice crystal growth, the reddish DMSO was only found within non-frozen regions, and the color intensity of the labeled DMSO gradually increased with ice crystal growth. As an alternative illustration, during ice melting, no reddish DMSO was initially observed near the newly melted ice crystals, and the average color intensity of the melting solution containing a DMSO–water mixture gradually decreased owing to the dilution of DMSO in the melting ice solution (Supplementary Video 2, Supplementary Fig. 2). Next, we quantified the changes in DMSO concentration in gelatin–GA precursor solution throughout the entire ice growing and melting process and found the same trend of DMSO enrichment to ~ 40% during ice growing (at −20 ℃) and the DMSO dilution during ice melting (Fig. 1c). We then directly visualized the regulatory effect of DMSO on the size of ice crystals formed in a DMSO–water mixture and determined that higher DMSO concentrations led to the formation of smaller ice crystals (Supplementary Video 3, 4, 5, Fig. 1d). Finally, we demonstrated the pore-controlling effect of DMSO on cryogelated gelatin 3D bioscaffolds upon freeze-drying using scanning electron microscopy, which showed a decrease in pore size from ~ 86 µm to ~ 26 µm when the DMSO concentration was increased from 0 to 10% (Fig. 1e). In addition, we found that DMSO played a similar role in the regulation of pore size in other porous scaffolds (e.g., gelatin-HA scaffold cross-linked by EDC (1-ethyl-3-(3-dimethylaminopropyl)-carbodiimide) (Supplementary Fig. 3) and that other cryoprotectants (e.g., glycerol and methanol) exhibited similar regulatory effects on scaffold pore sizes (Supplementary Fig. 4). Together, these results suggest a universal applicability of the cryoprotectant-enabled pore control method of producing porous bioscaffolds.

**Separate control of stiffness and pore size.** The stiffness of polymeric biomaterials can be readily controlled by varying the degree of chemical cross-linking through addition of chemical crosslinkers such as glutaraldehyde (GA), used in this study. However, changes in stiffness inevitably also affected the pore size of the resulting bioscaffold, with a lower GA concentration

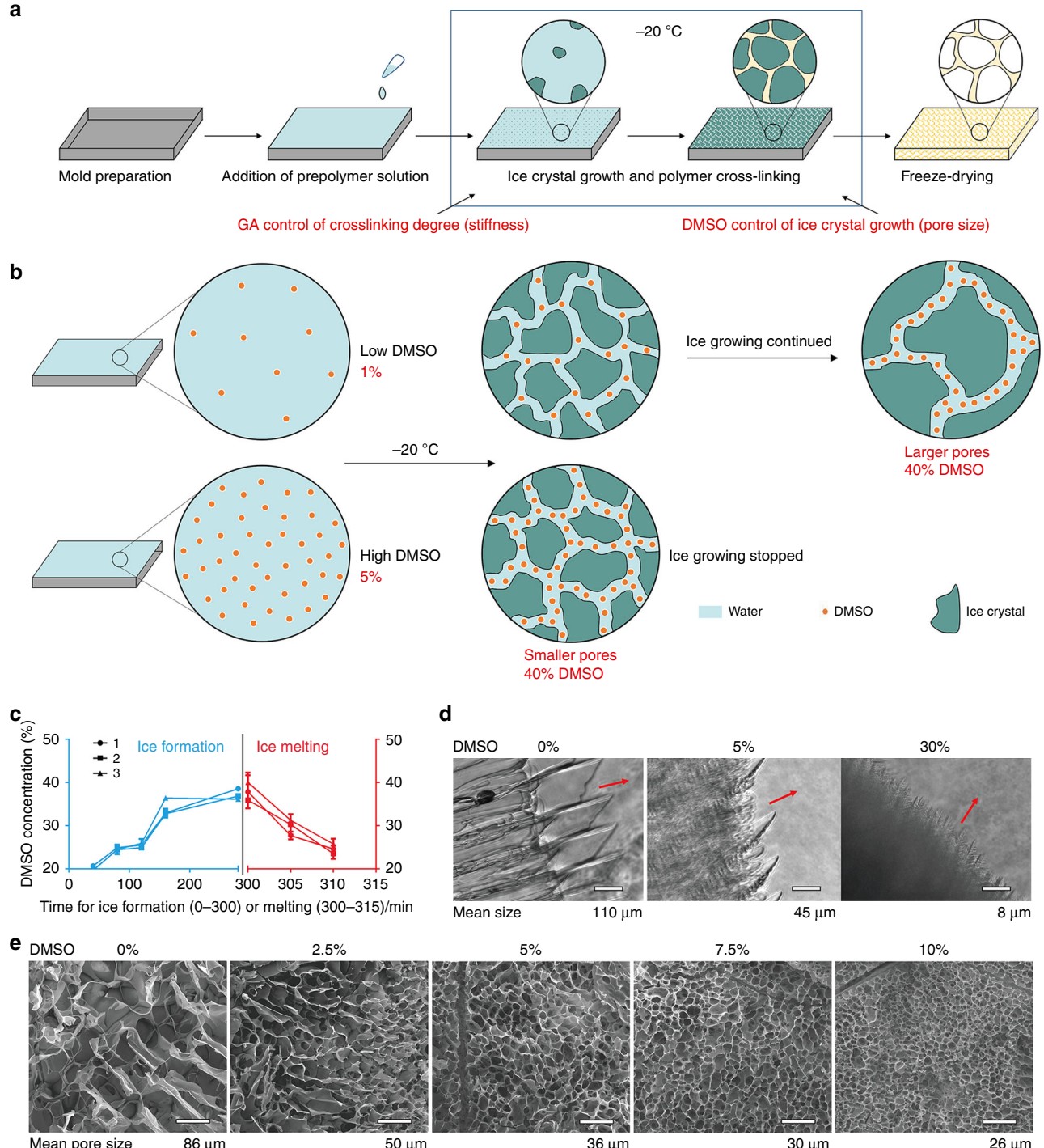

**Fig. 1** Precise control of scaffold pores sizes by cryoprotectant-regulated ice crystal growth. **a** Schematic of scaffold fabrication via cryogelation of pre-polymer solution (e.g., gelatin, HA, or PEG) in the presence of cryoprotectant (e.g., DMSO) for pore size control and chemical cross-linker (e.g., GA) for stiffness control. **b** Mechanism for DMSO-enabled pore size control via regulation of ice crystal growth. A higher initial concentration (e.g. 5% vs. 1%) of DMSO in the gelatin–GA precursor solution caused faster DMSO accumulation in the non-frozen liquid phase to a concentration of ~ 40%, at which point further ice crystal growth was inhibited, resulting in smaller ice crystals and corresponding pore sizes. **c** Characterization of DMSO accumulation during ice crystal growth in gelatin–GA precursor solution. Data are means ± s.e.m. ($n \geq 2$). **d** Ice crystal growth depends on the DMSO concentration in directional freezing (arrows indicate the direction of ice growth). Scale bar: 100 μm. **e** SEM images of gelatin scaffolds cross-linked by GA demonstrating the correlation of pore size and DMSO concentration. Scale bar: 100 μm

resulting in a larger pore size (Fig. 2a). The introduction of DMSO into the cryogelation system dramatically stabilized the scaffold pore sizes despite changes in scaffold stiffness. Specifically, when 1% (Fig. 2b) or 5% DMSO (Fig. 2c) was included in addition to variable concentrations of GA, the pore size of the gelatin scaffold remained ~ 60 μm or 30 μm, respectively, with the bulk Young's modulus within the range of 3 kPa to 20 kPa as measured by the tensile test. We further showed that varying the

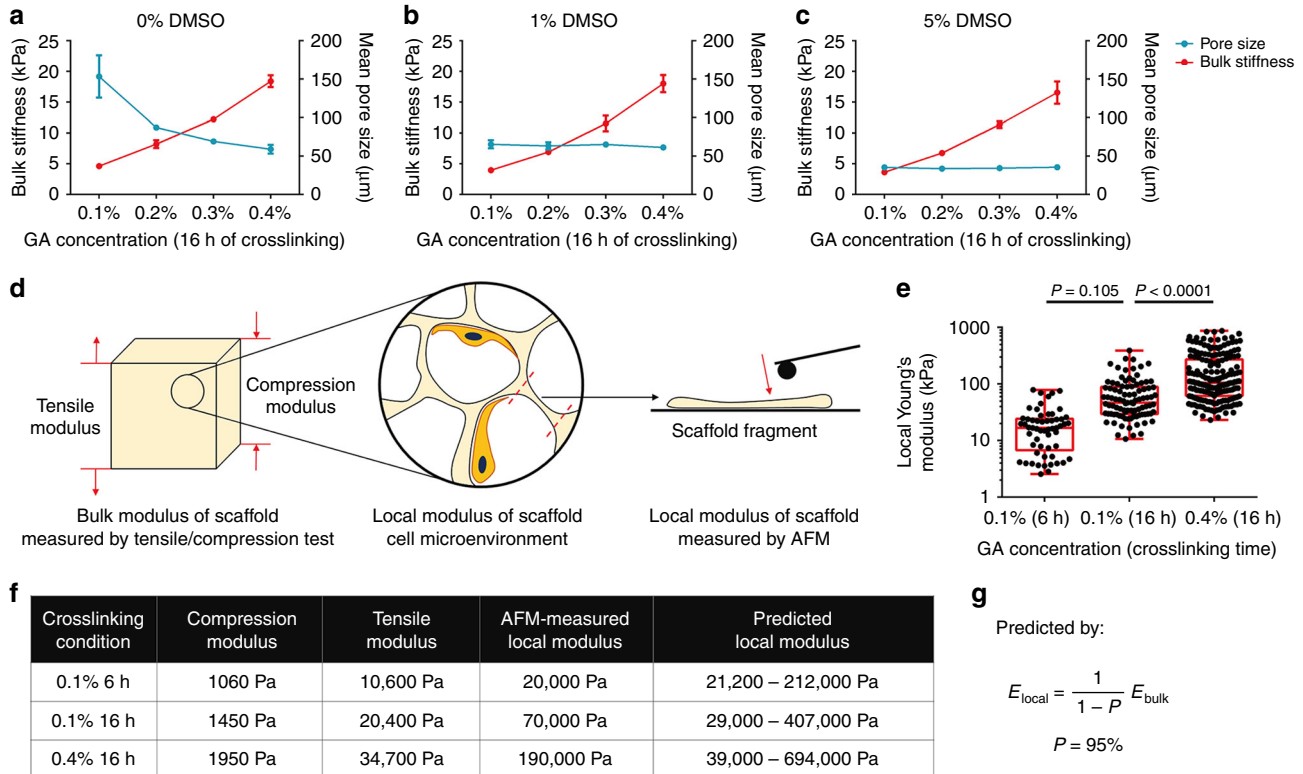

**Fig. 2** Independent control of scaffold stiffness and pore size. **a** Traditional stiffness control using varied concentrations of chemical cross-linker (GA) caused a change in scaffold pore size (stiffness, $n = 3$; pore size, $n = 5$). **b**, **c** With the introduction of 1% or 5% DMSO, the pore size remained unchanged at 30 μm or 60 μm, respectively, regardless of GA concentration (stiffness, $n = 3$; pore size, $n = 5$). Data are means ± s.d. **d** Schematic of the methods used to characterize the mechanical properties of porous scaffolds, including the bulk stiffness by the compressive/tensile test and the local stiffness by the AFM test of scaffold fragments. **e** Box and whisker plot of results from AFM test revealed a 10 times difference in local scaffold stiffness from the varied cross-linking conditions. (center line, median; box limits, upper and lower quartiles, whiskers, 1.5 × interquartile range) ($n \geq 57$, ANOVA). **f** Correlation between the bulk scaffold compressive/tensile modulus and the local scaffold modulus measured by AFM and predicted by the theoretical model. **g** Theoretical model for local modulus prediction

DMSO concentration within the range of 0–5% did not significantly alter the bulk scaffold stiffness (Supplementary Fig. 5a). These results implied the feasibility of adjusting the scaffold stiffness and pore size independently.

It should be noted that the stiffness measurements obtained above were based on tensile tests, which reflect the bulk mechanical properties of the scaffolds. As attached cells within porous scaffolds can only sense the local stiffness of the scaffold walls, we further measured the local stiffness of fragmented gelatin scaffolds via atomic force microscopy (AFM) (Fig. 2d, Supplementary Fig. 6), which can be regulated in the range of 20 kPa–190 kPa by varying the cross-linking conditions (Fig. 2e). Similarly, varying the DMSO concentration within the range of 0–5% did not significantly alter the local stiffness (Supplementary Fig. 5b). We next examined the correlation between the local and bulk stiffness of the scaffolds (Fig. 2f). A mathematical model was developed to predict the correlation between local and bulk stiffness (see methods, Fig. 2g, Supplementary Fig. 7). We determined this correlation to be solely dependent on the porosity ($\sim 95\%$ in this study) of the scaffold (Supplementary Fig. 8). The local modulus measured by AFM fell within the range of predicted values, which were calculated from the compressive modulus and tensile modulus (Supplementary Fig. 9). Together, these experimental and theoretical results demonstrated the feasibility of precise and separate control of the stiffness and pore size of 3D porous scaffolds.

**Mechano-responsiveness of fibroblasts in 3D porous scaffold.** Our systems enabled us to systematically investigate cell mechano-responsiveness in porous scaffolds, an area that has historically been challenging to pursue within the mechanobiology field. We began by investigating fibroblasts owing to their prevalence as the main cellular components for ECM construction and tissue remodeling during regeneration and disease. F-actin-stained human dermal fibroblasts (HDFs) exhibited elongated morphology within the large-pore (80 μm) scaffolds regardless of scaffold stiffness (Supplementary Video 6, 7), and cell spreading only occurred within small-pore (30 μm) scaffolds with relatively low local stiffness (20 kPa), but not in more rigid scaffolds (Supplementary Video 8, 9, Fig. 3a, b).

Next, the proliferation of cells inside porous scaffolds over the course of 5 days culturing was investigated. In general, scaffolds with larger and softer pores facilitated HDF proliferation compared with their smaller and more rigid counterparts (Fig. 3c), a phenomenon that might be explained by the greater availability of space and flexibility for cell growth and remodeling within scaffolds with large and soft pores.

Fibroblasts have been found to be sensitive to substrate stiffness in 2D culture, as demonstrated by the observation that stiff substrates such as normal plastic culture dishes induced the activation of fibroblasts to myofibroblastic phenotypes with upregulated α-smooth muscle actin (αSMA) expression[17,18]. Therefore, HDFs expressing a high level of αSMA were firstly primed on a soft 2D poly ethylene glycol (PEG) hydrogel

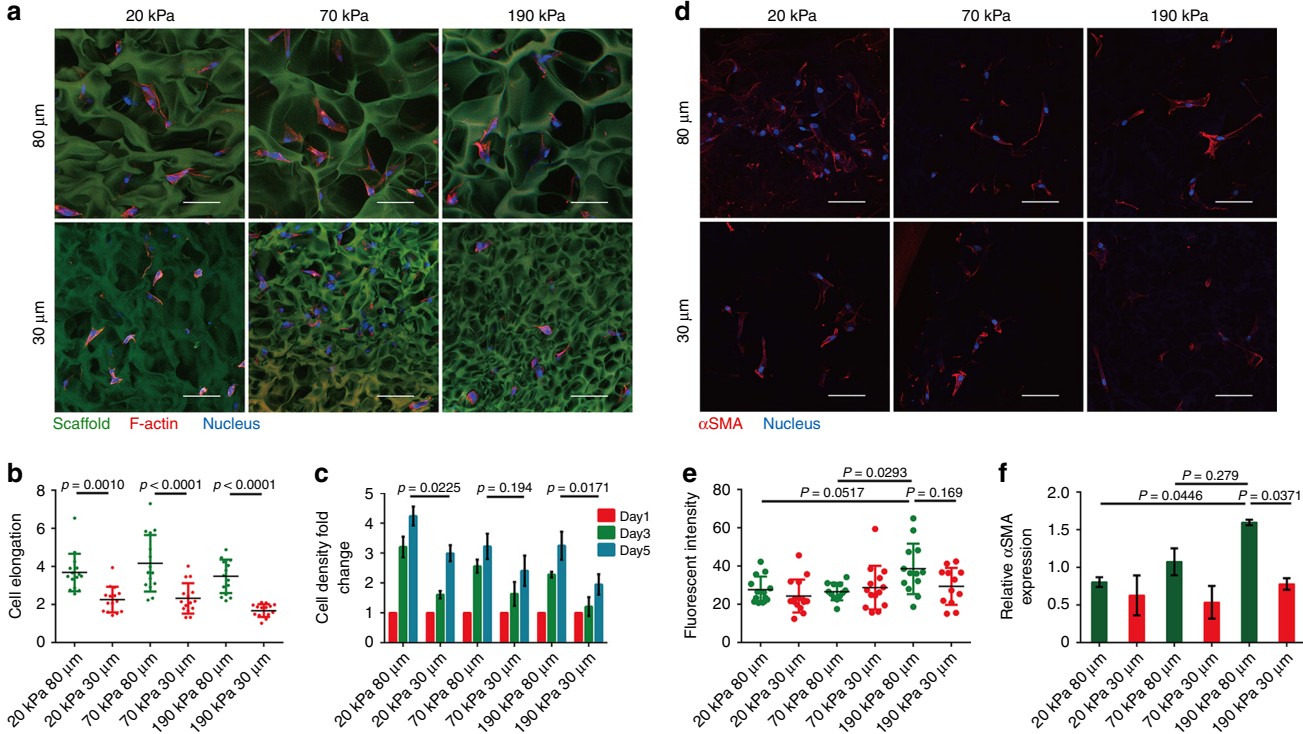

**Fig. 3** Fibroblasts respond to the stiffness and pore size of 3D porous scaffolds. **a, b** Confocal fluorescent imaging of F-actin-stained HDFs within auto-fluorescent scaffolds showing elongation of HDFs within large-pore scaffolds regardless of scaffold stiffness; whereas cell elongation only occurred in small-pore scaffolds with lower local stiffness (20 kPa) but not in stiffer scaffolds. Scale bar: 50 μm ($n = 15$). **c** Proliferation of HDFs inside the different porous scaffolds over 5 days of culturing ($n = 3$). **d, e** Significant αSMA expression with fiber-like accumulation was only observed in scaffolds with larger and stiffer pores (80 μm and 190 kPa). Scale bar: 50 μm. ($n \geq 12$). **f** Gene expression levels for αSMA ($n = 3$). Data are means ± s.d. (ANOVA)

substrate to be inactivated to a fibroblastic phenotype and then seeded into different 3D porous scaffolds (Supplementary Fig. 10). Only scaffolds with the highest stiffness and largest pore size (80 μm and 190 kPa) facilitated fibroblast activation, as characterized by intensified αSMA immunostaining and upregulated gene expression of *αSMA*. In contrast, scaffolds with softer and/or smaller pores maintained HDFs in a relatively inactive phenotype (Fig. 3d–f).

Next, we explored the expression of YAP, a key mechanotransduction transcription factor, whose nuclear localization upregulates αSMA expression in fibroblasts, as well as the activation of HDFs within 3D porous scaffolds[19]. As expected, fibroblasts in scaffolds with the highest stiffness and largest pores exhibited the highest nuclear-to-cytoplasm ratio for YAP (Supplementary Fig. 11a, b, c), which correlated well with αSMA expression. In addition, the scaffold-regulated expression of collagen I, one of the main ECM components secreted by activated fibroblasts, exhibited similar patterns as the YAP expression (Supplementary Fig. 11d). These data highlight the need for fine control of biophysical properties to specifically tailor cellular mechano-responsiveness in 3D porous scaffolds.

**Mechano-responsiveness of macrophages in 3D scaffolds.** We next examined the mechano-responsiveness of macrophages, which are key immunoregulatory cells that are polarized into a spectrum of states from pro-inflammatory (M1) to pro-healing (M2) phenotypes[20]. Previous studies have revealed the correlation between macrophage polarization and morphology[21,22]. Thus, we first studied the scaffold-regulated morphology of macrophages. As expected, a small-pore size tended to hinder the elongation of primary mouse bone marrow-derived macrophages (BMDMs)

inside the scaffold, whereas BMDMs exhibited elongated and stretched shapes in scaffolds with large pores (Fig. 4a, g). It should be noted that the morphology of macrophages seems to be mainly responsive to structural features but insensitive to scaffold stiffness. Next, polarization-related phenotypes were characterized through immunostaining for MHC-II, a marker associated with macrophage activation and initiation of the immune response. Surprisingly, a high ratio of MHC-II-positive BMDMs was observed in scaffolds with relatively soft (i.e., 20 kPa and 70 kPa) and small (i.e., 30 μm) pores, with over 50% of BMDMs showing an activated phenotype, but not in scaffolds with stiff (i.e., 190 kPa) and small (i.e., 30 μm) pores. Meanwhile, scaffolds with relatively stiffer and larger pores induced a significantly lower ratio of MHC-II expression, with only ~ 20% of macrophages exhibiting an activated phenotype (Fig. 4b, h). Flow cytometry analysis of MHC-II expression facilitated quantitative analysis of macrophage activation, which showed similar regulatory patterns that those ~ 50% MHC-II-positive BMDMs in scaffolds with soft and small pores (i.e., 20 kPa/70 kPa, 30 μm) and those only 12% MHC-II BMDMs in scaffolds with the high stiffness and large pores (i.e., 190 kPa, 80 μm) (Fig. 4e). We further characterized M1/M2 phenotype of macrophages using iNOS (M1)/Arginase-1 (M2). Over 60% of iNOS-positive cells could be observed in scaffolds with relatively soft (i.e., 20 kPa and 70 kPa) and small (i.e., 30 μm) pores, whereas only 30% could be observed in scaffolds with stiff and large pores (i.e., 190 kPa, 80 μm). In contrast, over 30% of Arginase-1-positive cells could be observed in scaffolds with stiff and large pores (i.e., 190 kPa, 80 μm), whereas < 10% could be observed in scaffolds with soft and small pores (i.e., 20 kPa, 30 μm) (Fig. 4c, d, i, j). Besides, flow cytometry analysis also validated that macrophages tended to be M1 phenotype in scaffolds with soft and small pores (i.e., 20 kPa, 30 μm)

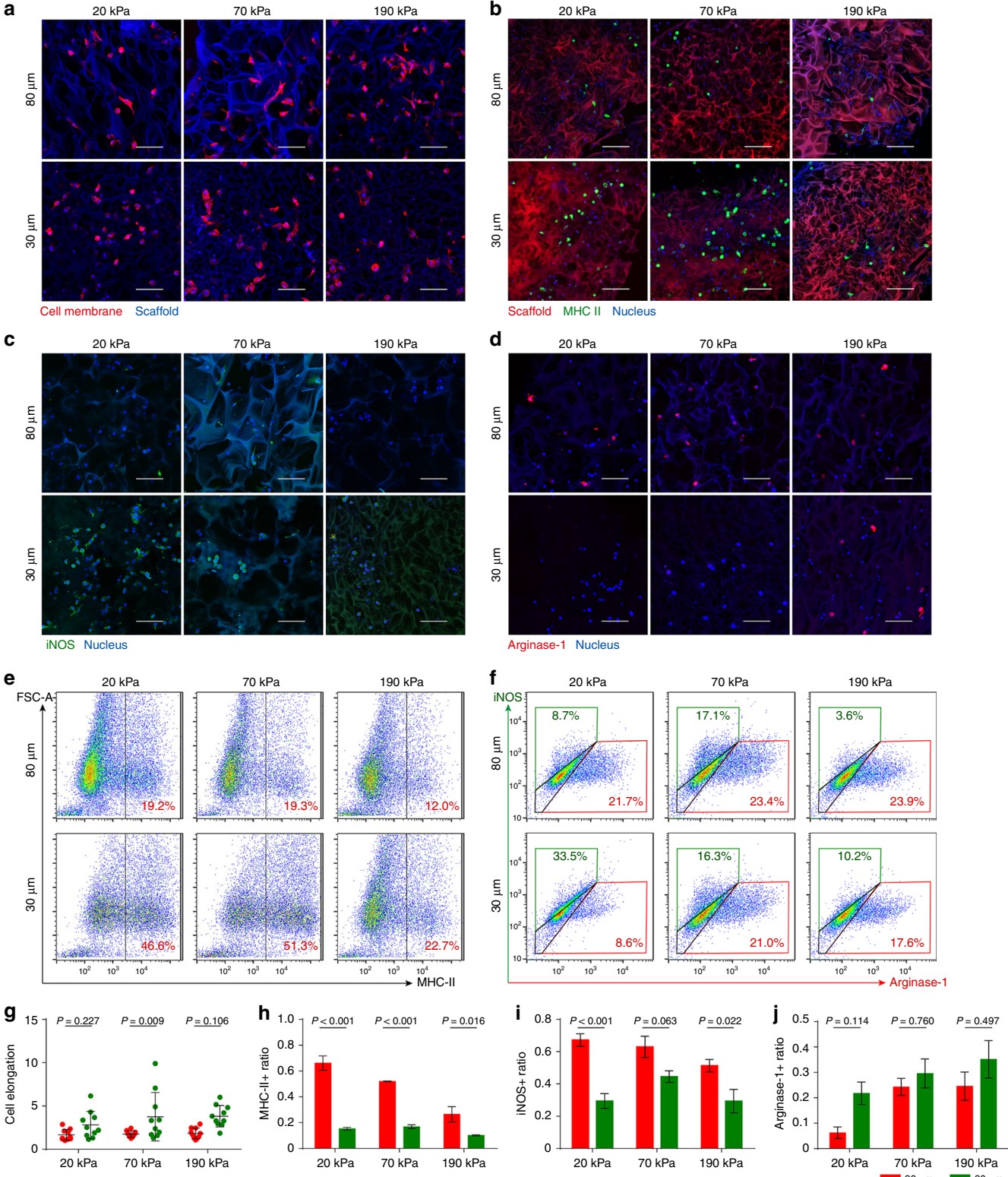

**Fig. 4** BMDM phenotypes can be modulated by pore size and stiffness of the scaffolds. **a**, **b** Fluorescent images of BMDMs stained for plasma membrane (red) and MHC-II (green) in auto-fluorescent gelatin 3D porous scaffolds. **c**, **d** Fluorescent images of BMDMs stained for iNOS (M1, green) and Arginase-1 (M2, red) in auto-fluorescent gelatin 3D porous scaffolds. **e** Flow cytometry analysis of MHC-II-positive cells in scaffolds with different pore sizes and stiffness. **f** Flow cytometry analysis of iNOS/Arginase-1-positive cells in scaffolds with different pore sizes and stiffness. **g–j** Quantification of cell elongation ($n = 10$), the ratio of MHC-II-positive cells ($n = 3$), iNOS-positive cells ($n = 4$) and Arginase-1-positve cells ($n \geq 3$) in scaffolds with different pore sizes and stiffness in fluorescent images. Scale bar: 50 μm. Data are means ± s.e.m. (ANOVA)

(33.5% iNOS + / 8.6% Arginase-1 +) and tended to be M2 phenotype in scaffolds with stiff and large pores (i.e.,190 kPa, 80 µm) (3.6% iNOS + / 23.9% Arginse-1 +) (Fig. 4f). To confirm the variances in macrophage phenotypes, gene expression of a representative cytokine panel (i.e., pro-inflammatory cytokines *Il-6*, *Il-1β*, and *Tnfα* and anti-inflammatory cytokine *Il-10*) were characterized by qPCR, which was nicely correlated with iNOS/Argianse1 expression. Specifically, higher expression of *Il-6*, *Il-1β*, and *Tnfα* and lower expression of *Il-10* was only observed for BMDMs within scaffolds with soft and small pores (Supplementary Fig. 12).

Finally, the regulatory effect of scaffolds' physical properties was evaluated in the presence of additional immunoregulatory soluble factors (i.e., 50 ng ml$^{-1}$ LPS for M1 induction, 20 ng ml$^{-1}$ IL-4 + 20 ng ml$^{-1}$ IL-13 for M2 induction). When BMDM were cultured in scaffolds with additional M1 induction, overall increase in iNOS level and decrease in Arginase-1 level were observed with high-percentage iNOS + / low-percentage Arginase-1 + cells within soft and small (i.e., 20 kPa, 30 µm) pores and low-percentage iNOS + / high-percentage Arginase-1 + cells within stiff and large pores (i.e., 190 kPa, 80 µm) (Supplementary Fig. 13). When BMDM were cultured in scaffolds with additional M2 induction, sharp decrease in iNOS level and increase in Arginase-1 level were observed for all groups with the lowest percentage of Arginase-1 + cells within soft and small (i.e., 20 kPa, 30 µm) pores reaffirming the tendency of pro-inflammatory induction by the scaffolds with small pores (Supplementary Fig. 14).

In sum, our study provided the evidence that physical properties of the scaffolds, namely the pore stiffness and pore size, could regulate macrophage phenotype. Specifically, macrophages in scaffolds with small and soft pores tend to be activated towards pro-inflammatory phenotype, whereas cells in scaffolds with larger and stiffer pores are prone to anti-inflammatory phenotype activation. The regulation on macrophages by the physical properties of the scaffolds is still prominent even in the presence of additional immunoregulatory soluble factors.

**Physical confinement induces cell responses within small pores.** Till now, our study has highlighted the key role played by scaffold pore size in regulating the cellular behaviors of both HDFs and BMDMs. Next, we naturally sought to determine how the structural features of porous scaffolds regulate cellular responses. As illustrated above, pore sizes may impose different degrees of physical confinement to the cells, which can be expected to influence cell morphology and function. To validate this assumption, fluidic osmotic pressures were applied by the introduction of inert molecules (e.g., PEG) into the culture medium to simulate the physical confinement of cells, as described in a previous study[23]. Specifically, 0–12% PEG 400 was added to the culture medium to generate a gradient of osmotic pressure in which the HDFs or BMDMs were cultured either in a conventional 2D culture dish or a 3D gelatin porous scaffold. With increasing osmotic pressure, the cell spreading of 2D-cultured HDFs and BMDMs was gradually hindered, with more rounded cells evident under high osmotic pressure (e.g., 8% and 12% PEG) (Fig. 5a, c; Fig. 5b, e), which generated different degrees of physical confinement. Interestingly, phenotypic markers in 2D-cultured HDFs and BMDMs, namely, αSMA and MHC-II, exhibited marked changes in response to changes in osmotic pressure. For αSMA expression in HDFs, higher osmotic pressure resulted in reduced αSMA expression (Fig. 5d), in line with the reduction of αSMA observed in HDFs within scaffolds with small pores. Moreover, high osmotic pressure was also shown to induce the nuclear export of YAP (Supplementary Fig. 15), indicating

that HDFs can sense osmotic pressure-induced physical confinement through a mechanically relevant signaling pathway. Meanwhile, as the osmotic pressure increased, the ratio of MHC-II-positive BMDMs increased from 5 to 50% (Fig. 5f) alongside an increase in pro-inflammatory cytokine (i.e., *Il-6*, *Il-1β*, and *Tnf-α*) secretion and a decrease in anti-inflammatory cytokine (i.e., *Il-10*) secretion (Supplementary Fig. 16). These results indicate that osmotic pressure-induced physical confinement alone can induce a pro-inflammatory phenotype in macrophages.

We next examined whether osmotic pressure-induced physical confinement influences cellular phenotypes in HDFs and BMDMs cultured in 3D gelatin scaffolds with large pore sizes (i.e., 80 µm). As a result, higher osmotic pressures markedly hindered the cellular spreading of both HDFs (Fig. 5g, i) and BMDMs (Fig. 5h, k) even in scaffolds with the highest pore size and local stiffness. Meanwhile, stiffness-dependent HDF activation, as represented by αSMA upregulation, was completely blocked in scaffolds with large pores under high osmotic pressure (i.e., 8% PEG) (Fig. 5j), similar to the regulatory effect of small-pore sizes (Fig. 3e). More interestingly, pro-inflammatory polarization of BMDMs, as represented by MHC-II upregulation, was realized in scaffolds with large and soft pores under high osmotic pressure (Fig. 5l), whereas under normal conditions, BMDM polarization was only achievable in scaffolds with small and soft pores (Fig. 4d). These results demonstrate the deterministic role played by physical confinement in the regulation of cellular phenotypes in 3D porous scaffolds.

**In vivo cellular responses to implanted 3D porous scaffolds.** To explore in vivo cellular responses to 3D porous scaffolds and their potential applications, we subcutaneously implanted gelatin scaffolds with various pore sizes and stiffness in a mouse wound healing model. As the implanted bioscaffolds were expected to regulate endogenous cell infiltration and function, we first tested whether scaffolds with a smaller pore size hindered cell infiltration both in vitro and in vivo (Fig. 6a). When HDFs were seeded on top of the porous scaffolds in vitro, most cells could not reach a depth of 400 µm in the scaffold with 30 µm pores, whereas cell infiltration was not disturbed in the scaffolds with 80 µm pores (Fig. 6b, d). In parallel, endogenous cell infiltration in vivo was investigated after subcutaneous implantation of scaffolds. No significant cell infiltration was observed two days after implantation in all implanted scaffolds (Supplementary Fig. 17a). Four days after implantation, a large number of cells were found to be accumulated within scaffolds with 80 µm pores, whereas scaffolds with 30 µm pores contained significantly fewer cells (Fig. 6c, e).

To further characterize the infiltrated cells inside the scaffolds, immunostaining of αSMA and MHC-II/CD11b was performed to analyze the activation state of stromal fibroblasts and macrophages. Few αSMA-positive cells were found within the scaffolds, indicating relatively rare infiltration of myofibroblastic cells or other pericytes into the scaffolds (Supplementary Fig. 17b). Meanwhile, abundant MHC-II/CD11b-positive macrophages were detected within the scaffolds with soft (20 kPa) and small (30 µm) pores, but not in the other groups (Fig. 6f, h). The specific accumulation of pro-inflammatory macrophages in the scaffold with soft and small pores was further confirmed by flow cytometry analysis of cells retrieved from the implanted scaffolds (Fig. 6g, i). The in vivo scaffold regulation of macrophage polarization was surprisingly consistent with the conclusion drawn in the in vitro study that scaffolds with relatively softer (20 kPa) and smaller (30 µm) pores tend to induce the pro-inflammatory polarization of macrophages. Eight days post implantation, wound sizes in all scaffold-implanted groups were significantly reduced and exhibited regenerated skin structures

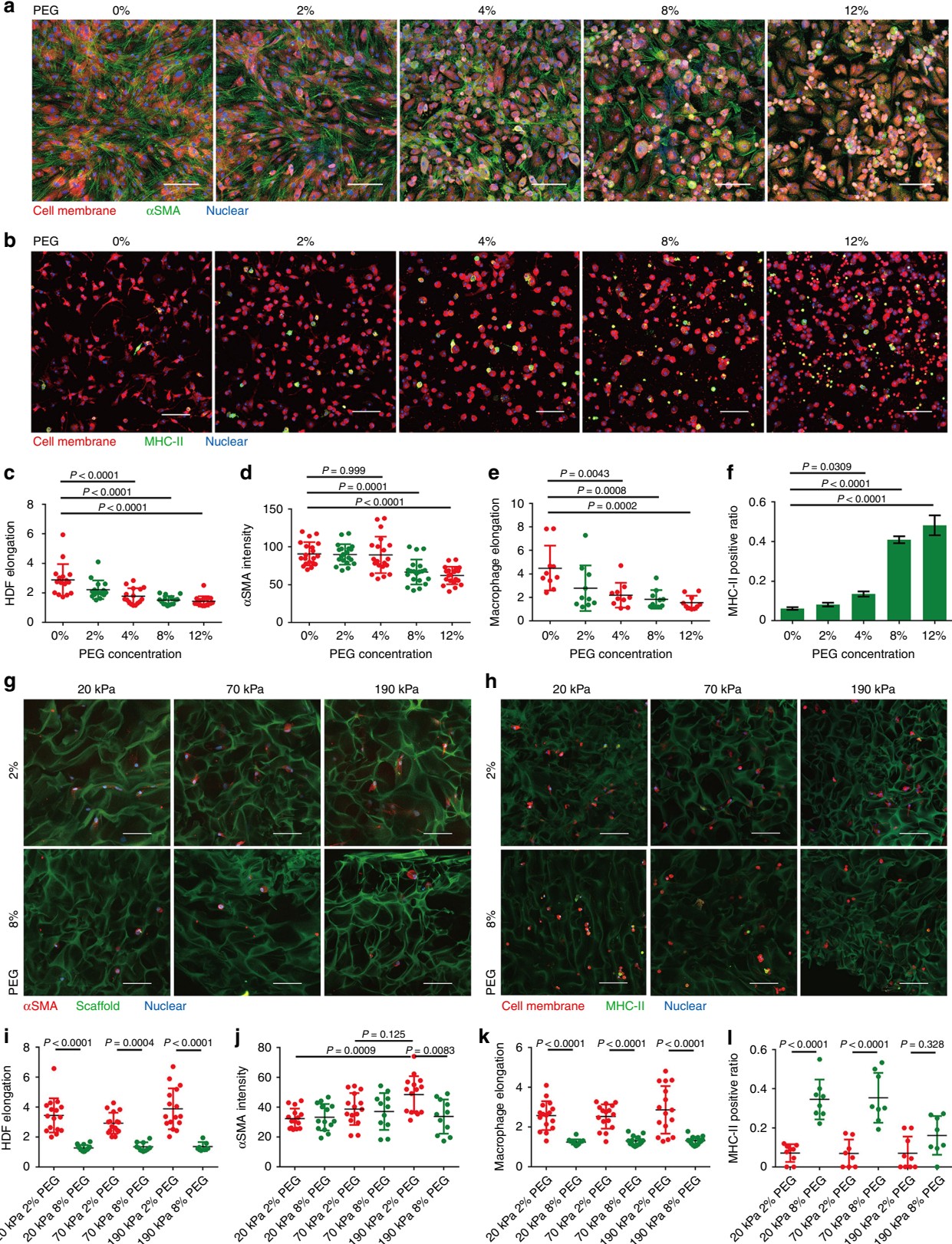

and accessories typically seen in the normal skin (Supplementary Fig. 18). Scaffolds with different biophysical properties seemed to influence the healing rate, with relatively larger (80 μm) and softer (20 kPa) pores accelerating wound closure more prominently, especially at the early stage (day 2–4).

The cell infiltration and foreign body responses induced by the implanted scaffolds were further evaluated in a mouse sub-cutaneous model (Supplementary Fig. 19). The greatest number of macrophages (F4/80) and neutrophils (Ly6G) were found to infiltrate into scaffolds with small and soft pores (i.e., 20 kPa,

**Fig. 5** Physical confinement contributes to pore-regulated cellular mechano-responsiveness. **a**, **b** Confocal images of HDFs and BMDMs cultured on 2D substrates and subjected to physical confinement induced by osmotic pressure (tuned by PEG concentration). **c**, **d** Characterization of 2D-cultured HDF elongation ($n = 16$) and αSMA expression ($n = 20$) upon being subjected to different osmotic pressures. **e**, **f** Characterization of 2D-cultured BMDM elongation ($n = 10$) and MHC-II expression ($n = 3$) upon being subjected to different osmotic pressures. **g**, **i**, **j** Characterization of osmotic pressure-induced HDF elongation ($n \geq 7$) and αSMA expression ($n \geq 11$) in 3D scaffolds with a large pore size and differing stiffness. **h**, **k**, **l** Characterization of osmotic pressure-induced BMDM elongation ($n = 16$) and MHC-II expression ($n \geq 8$) in 3D scaffolds with a large pore size and differing stiffness. Scale bar: 50 μm. Data are means ± s.d. (ANOVA)

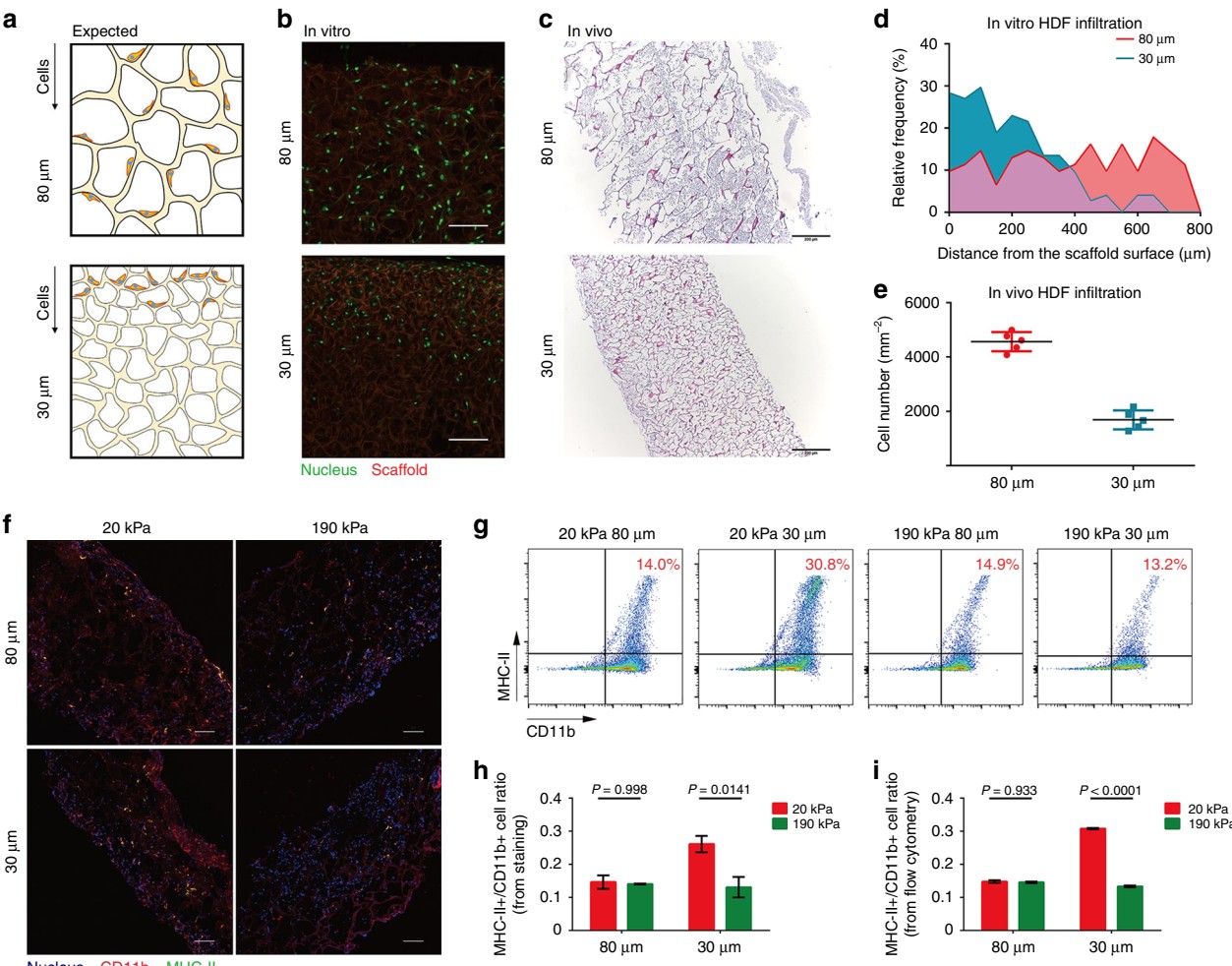

**Fig. 6** Physical properties of implanted scaffolds regulate in vivo cellular behavior. **a–c** Schematic illustration of resistance to cell infiltration in scaffolds with a smaller pore size compared with those with larger pores **a**, which was also verified by in vitro **b**, **d**, and in vivo **c**, **e** experiments (4 days after implantation). Scale bar: **b** 100 μm; **c**, 200 μm. **f** Staining of CD11b and MHC-II for infiltrated cells in different scaffolds, with quantification of the ratio of MHC-II-positive cells (4 days after implantation). Scale bar: 100 μm. **g** Cells inside the scaffolds were sorted to examine their CD11b and MHC-II expression to verify their cell fate. **h**, **i** Quantitative results of fluorescent images and flow cytometry showed highly consistent results for macrophage cell populations in vivo wherein smaller and softer pores resulted in a dramatic increase in MHC-II-positive cells ($n = 3$). Data are means ± s.e.m. (ANOVA)

30 μm) on Day 4, whereas, highest level of collagen deposition was found in scaffolds with large and stiff pores (i.e., 190 kPa, 80 μm) on Day 8. These results indicated that scaffolds with small and soft pores (i.e., 20 kPa, 30 μm) could induce pro-inflammatory response shortly after implantation. In contrast, scaffolds with large and stiff pores (i.e., 190 kPa, 80 μm) would reduce inflammatory response in early stage but may induce fibrotic responses later to exert the pro-healing effect. Our results indicated that by adjusting the physical properties of the implanted porous 3D scaffolds, we could regulate in vivo cellular response both in a skin wound healing model and a subcutaneous model. Scaffolds with small and soft pores will induce high level of inflammatory response and scaffolds with large and stiff pores

will promote fibrotic response and wound healing. Although there was no significant difference in wound healing rate among different groups, tuning the scaffolds' physical properties could potentially be leveraged to modulate foreign body response and tissue regeneration.

## Discussion

This study mainly focused on the modulation of the physical properties of 3D porous scaffolds and demonstrated the separate control of pore size and stiffness, providing a deeper mechanistic understanding of how cells sense the physical properties of porous scaffolds. Conventionally, the structural features of porous bioscaffolds are linked to their stiffness, making it difficult to

attribute cellular mechano-responsiveness to any single regulatory factor. By taking advantage of cryoprotectant-enabled pore size control, we decoupled pore size regulation from scaffold stiffness and achieved precise control of pore size to produce pores ranging from 15 μm to 100 μm. In addition to the cryoprotectant concentration, the reaction temperature and cooling rate of the cryogelation process also affected ice crystal formation and pore size thereafter. Specifically, the cooling rate of the cryogelation process could be regulated by altering the size of the mold, with a smaller mold resulting in faster freezing, smaller ice crystals and thus smaller pore sizes down to 15 μm (Supplementary Fig. 20). It should be noted that current approaches for cooling rate control are still limited owing to the inefficient heat transfer by air cooling. Further improvement can be achieved by using more efficient media for heat transfer (e.g., cryogelation in liquid freezing media instead of the current air cooling) and temperature controlling instrument for gradient cooling. It was notable that pore structure was stable, which remained unchanged after washing, refreezing, and lyophilization process (Supplementary Fig. 21). Meanwhile, the pore regulatory effect of DMSO was dependent on the reaction temperature during cryogelation and was most effective at relatively higher reaction temperatures (e.g., higher than −30°C) (Supplementary Fig. 22). It should be noted that several previous studies have supplemented DMSO as a component in the pre-polymer solution for cryogel preparation, which acted as a solvent to dissolve other substances such as PNIPAAm[24], PLGA[25] and N-vinylcaprolactam[26] for scaffold fabrication. Our study represents the first attempt of applying DMSO to control the pore size of the cryogel. Furthermore, the mechanism for DMSO-based pore size control is proposed and verified in our work.

The facile approach for controlling pore size developed in this study allowed us to delineate the independent effect of changes in pore size on cellular mechano-responsiveness, which was previously unachievable. Specifically, the physical confinement of cells cultured within scaffolds with smaller pore sizes resulted in the inhibition of cell elongation and the regulation of specific genetic and epigenetic signaling pathways, as recently reported[27]. This regulation ultimately induced the inactivation of fibroblasts and promoted a pro-inflammatory phenotype for macrophages. And it is notable that the regulation on macrophages by the physical properties of the scaffolds is prominent even in the presence of additional immunoregulatory soluble factors. We briefly demonstrated that the downstream regulatory mechanism of fibroblasts was related to YAP, a well-known mechanotransductive transcription factor. Deep and systematic mechanistic study will be required to reveal more genetic/epigenetic regulators of this process. Though we identified pore size as a key regulator that imposes physical confinement of cells within porous scaffolds, pore size also affected mass transfer and cell–cell communication. In this study, we used low cell density to reduce the influence of cell–cell communication within the scaffolds. Meanwhile, cells were viable in all culture conditions. There was no significant difference in viability between cells in the center and edges of all scaffolds, excluding variances in mass transfer among different scaffolds.

It should be noted that both the stiffness and pore size of scaffolds may be relevant to the biochemical properties of the scaffolds such as the ligand density on the scaffold surface, which are also important regulators of cell behavior. Therefore, biochemical influences must be considered, whereas investigating the effects of biophysical cues on cellular mechano-responsiveness. In this study, we do not expect a significant influence of the biochemical properties on biophysical investigation owing to the variance in ligand densities of the gelatin scaffolds, which are mainly RGD motifs[28]. The RGD surface density of the various

gelatin scaffolds studied here was estimated (see methods) to exhibit an average gap of 5 nm between each RGD motif. A gap of < 100 nm between each RGD motif was previously proven to be sufficient for activation of focal adhesion formation to mediate effective cell adhesion on substrates with stiffness ranges similar to those of the gelatin scaffolds described in this study[29]. In addition, GA residue was not detectable using HPLC modified from the protocol recommended by the United States Pharmacopeia, and endotoxin contents of the scaffolds showed low level ranged from 0.1 to 0.2 EU mg$^{-1}$ (Supplementary fig. 23). Thus, it is reasonable to assume that the responses of the cells in this study mainly result from changes in the physical properties of the scaffolds, whereas the influence of the varying biochemical properties is negligible.

The local stiffness of the scaffolds as measured by AFM has been adopted as a biophysical property reflecting the mechanical features that a cell can sense within the scaffold. Cells can spread over several interconnected pores within the scaffolds, thus bending the pore wall and remodeling local stiffness. Therefore, the cells inside the porous scaffolds were expected to experience significant heterogeneity in local stiffness[30]. High-resolution imaging tools and single cell analysis may be adopted to explore the heterogeneity issue in future investigation.

Our invention not only provides powerful tools for mechanobiological study but also offers insights for the optimization of implantable scaffolds to improve regenerative therapy in a clinical setting. Porous scaffolds are implanted biomaterials widely applied in tissue engineering and regenerative medicine. Great efforts have been made to improve their therapeutic efficacy through the chemical modification of bioscaffolds with bioactive factors such as growth factors. However, the high cost and fast proteolytic degradation limited the wide application of chemically modified bioscaffolds. In addition, side effects may be induced by uncontrolled release and diffusion of bioactive molecules into peripheral wound tissues, which make it difficult for clinical use[31]. Therefore, developing novel biomaterials with distinct biomechanical properties, which could regulate cell behavior and host response, has emerged as a promising strategy. Our study clearly demonstrated the importance of the structural and mechanical features of bioscaffolds to the regulation of endogenous infiltration and phenotype transition as well as tissue regeneration, indicating that biophysical features should be considered in bioscaffold optimization for regenerative applications without altering the biochemical composition in bioscaffolds. Moreover, the concept and methodology developed in this study can also be applied to tailor the physical properties of scaffolds according to variations in the etiology and location of an injury to ultimately develop precise medical treatments for individual patients.

## Methods

**Animal experiments**. All animal experiments were kept to a strict protocol approved by the Animal Ethics Committee of the Center of Biomedical Analysis (IACUC),Tsinghua University, which is accredited by AAALAC (Association for Assessment and Accreditation of Laboratory Animal Care International).

**Materials and reagents**. Gelatin was obtained from Sigma-Aldrich (cold water fish skin, Cas. 9000-70-8, MW ~ 60 kDa), whereas the hyaluronic acid was obtained from Bloomage Freda Biopharm (MW = 1500 kDa ~ 7500 kDa). Other reagents included glutaraldehyde (Macklin), 1-(3-Dimethylaminopropyl)-3-ethylcarbodiimide hydrochloride (Aladdin), Cryoprotectant DMSO (SolarBio) and Osmotic pressure regulator PEG 400 (Sigma-Aldrich). Cell culture reagents included Dulbecco's Modified Eagle Medium (DMEM), Penicillin/Streptomycin and fetal bovine serum (FBS) (Wisent).

**Cryoprotectant-regulated preparation of porous scaffolds**. Preparation of gelatin-based porous scaffolds regulated with cryoprotectant was modified based on the established method[32]. In brief, 4% w/v gelatin precursor solution was

prepared with gelatin powder dissolved in deionized water, which was mixed with varied concentrations of cryoprotectant (0–5% v/v) such as DMSO, methanol and glycerol. Then, the pre-mixed solution was incubated on ice for 5 min. Varied concentration of glutaraldehyde (i.e., 0.1–0.4% w/v) was added then and the solution was stirred for 20 s. After that, a $20 \times 20 \times 1$ (mm)$^3$ poly(methyl methacrylate) (PMMA) mold was filled with 400 µl of the solution before it was placed in $-20\,°C$ refrigerator and underwent cryogelation for 16 h.

To prepare for gelatin-hyaluronic acid scaffold[33], the gelatin precursor solution was replaced with 2% gelatin solution mixed with 0.5% 1-(3-Dimethylaminopropyl)-3-ethylcarbodiimide hydrochloride activated 0.5% hyaluronic acid. After varied concentration of cryoprotectants was added to the precursor solution, it was straightly placed in $-20\,°C$ refrigerator and underwent cryogelation for 16 h.

After 16 h of cryogelation, the scaffold was placed in room temperature to allow ice melting and it was carefully washed with deionized water to remove residual reagent. For gelatin scaffold, 1% sodium borohydride solution (pH = 9.7) was introduced to remove residual glutaraldehyde for 10 min, followed by extensive washing with deionized water. The well-cleaned scaffolds were collected in dishes and placed in $-20\,°C$ for 2 h then lyophilized for 2 h. Finally, the scaffolds were collected and stored in low vacuum environment for the following characterization, cell culture, and implantation.

**Pore size determination**. For each cryoprotectant concentration, three replicates of scaffolds were collected. For imaging, three $1 \times 1 \times 1$ (mm)$^3$ pieces of scaffolds were cut from the origin scaffold in random location. High resolution images of these pieces were acquired using scanning electron microscope. Three images (magnification $\times 1000$) were recorded for each piece of scaffold. We measured the long and short axes of each pore by ImageJ and took the average as the final diameter of that pore. Then 20 pores were randomly selected in each image and 9 images were used for each group to acquire the final results.

**Ice crystal growing assay**. A $25 \times 75$ (mm)$^2$ glass-slide was subjected to Plasma treatment (PDC-32G, Harrick Plasma, USA) for 1 min to increase the hydrophilicity. A thin liquid film then could form on the surface when 50 µl water or cryoprotectant-water mixture was added on the surface of slide. Hydrophobic dye such as oil red was introduced to indicate the existence of cryoprotectant. Meanwhile, a $20 \times 20 \times 2$ (mm)$^3$ PMMA mold was pre-cooled in the liquid nitrogen. To begin the experiment, the slide was placed under a light microscope to image the water film. Then the pre-cooled PMMA mold was quickly moved from the liquid nitrogen to the surface of slide to induce the ice crystal formation and growth. The whole process could be recorded under the microscope.

**DMSO concentration measurement**. During the ice formation process, the solid and liquid phase of the ice-water/DMSO mixture was separated. In briefly, 30 ml of mixture was centrifuged in $-20\,°C$ for 5 min, then the liquid phase of the mixture would accumulate at the bottom due to the higher density, which was easily separated by pipetting. To quantify the DMSO concentration, a standard curve about the density and the DMSO concentration of water–DMSO mixture was firstly acquired. The separated liquid phase containing DMSO–water mixture was allowed to recover to room temperature before the measurement of its density, which could be calculated according to the standard curve.

**Tensile/compression test for bulk stiffness**. For preliminary evaluation of the bulk stiffness influenced by GA in Fig. 2a–c and supplementary fig. 5a, scaffolds with $100 \times 50 \times 2$ (mm)$^3$ shape were stretched by Force digital gauge (Haibao Instrument), and bulk stiffness was estimated by force/strain at 80% strain. For tensile test in Fig. 2d, f and supplementary fig. 9, the scaffold was shaped in a $20 \times 10 \times 2$ (mm)$^3$ mold and fixed on a mechanical measurement platform (QuantumScope) through two clamps and along the long axis of the scaffold. The platform provided a precise control of two clamps to realize the stretching of the scaffold, whereas a force sensor downside the lower clamp could record the tensile force with accuracy of 0.0001 N and sampling time of 0.01 s. During the tensile test, the origin length of the scaffold was measured before the stretching. And the stretching progress was carried on until the scaffold broke apart. Thus a force–displacement curve could be acquired. Since we already knew the size of the scaffold (origin length $L_0$; Section area $S$), the tensile stress and strain could be calculated according to equation (1) (2):

$$Force/S = Tensile\ Stress \qquad (1)$$

$$Displacement/L_0 = Tensile\ Strain \qquad (2)$$

And a stress–strain curve could be acquired to calculate the bulk tensile modulus of the scaffold.

For compression test, the scaffold was shaped in a $10 \times 10 \times 2$ (mm)$^3$ mold and placed on the same mechanical measurement platform as the tensile test. The test was performed with a column-shape probe. The radius of its cross section is 20 mm, which is much larger than samples. Compression force was recorded during the entire compression process, whereas the thickness of scaffold was measured

with 1.5 mm of displacement. Thus force–displacement curve and stress–strain curve could be acquired to calculate the compression modulus of the scaffold.

**AFM-based test for local stiffness measurement**. To allow the measurement of local stiffness for porous scaffold in AFM, a uniform and smooth surface was required. Then the scaffold was broken into debris using a grinder before the test. These small debris of scaffold should be small enough (smaller than 80 µm each) that single layer of scaffold wall could be observable under microscope. And these small debris of scaffold were attached on a cover slip to carry on the AFM test.

The AFM tip was modified before the test in which a small bead (radius 3 µm) was fixed on the terminal of the tip. During the test, the tip was controlled to approach the debris with a speed of 10 µm/s, until the bead reached the debris and a force over 1.5 pN was generated between the bead and the debris. The tip would be controlled to detach from the debris to finish a single test for local stiffness. Force–displacement curve was recorded during the whole process from which the local young's modulus could be calculated.

Meanwhile, the thickness of each debris was measured using AFM as a quality-control of the debris. In brief, after the measurement of stiffness for a single piece of debris, the tip was horizontally moved to a nearby region. The same measurement was then carried out to record the force–displacement curve. As the height of debris surface could be acquired in the former curve, whereas the height of cover slip surface could be acquired in the later curve. The thickness of debris could be calculated. AFM test for each scaffold included at least results of 20 different debris from the scaffolds, and each debris was measured for three times. Such experimental design gave us over 60 stiffness value for each group in every experiment, resulting in over 180 data point for each group in total.

**Calculation of the bulk/local stiffness ratio**. A hexagonal model was developed as illustrated in supplementary fig. 9. If a total force $\mathbf{F}$ is vertically applied on the scaffold and there is $m$ fibers in the same horizontal plane, the local force $\mathbf{f}$ applied on each single fiber should be:

$$\mathbf{f} = \frac{\mathbf{F}}{m} \qquad (3)$$

If the length $l$ and width $d$ of each fiber is known, the porosity ($P$) can be calculated by dividing the area of fibers ($S_{pore}$) with the total area of the scaffold ($S_{total}$):

$$P = \frac{S_{pore}}{S_{total}} = \frac{S_{total} - S_{fiber}}{S_{total}} = \frac{3\sqrt{3}l^2 - 3dl}{3\sqrt{3}l^2} = 1 - \frac{d}{\sqrt{3}l} \qquad (4)$$

The overlapping of fiber area can be ignored owing to the high porosity of the scaffolds. Moreover, the bulk stiffness $E_{bulk}$ can be acquired through the stress (which is determined by the total forces $\mathbf{F}$ divided by the width of scaffold $L$) and the overall strain $\varepsilon_{bulk}$. Although the local stiffness $E_{local}$ can be acquired through the stress (which is determined by the forces on each fiber $\mathbf{f}$ divided by the width of fiber $d$) and the strain of fibers $\varepsilon_{local}$:

$$E_{bulk} = \frac{\mathbf{F}/L}{\varepsilon_{bulk}} \qquad (5)$$

$$E_{local} = \frac{\mathbf{f}/d}{\varepsilon_{local}} \qquad (6)$$

When the strain is low, the fiber deformation could be ignored. Therefore, the bulk strain $\varepsilon_{bulk}$ should be equal to $\varepsilon_{local}$. Thus, the relationship of bulk stiffness and local stiffness can be represented in:

$$E_{local} = \frac{\mathbf{f}/d}{\mathbf{F}/L}E_{bulk} = \frac{L}{md}E_{bulk} \qquad (7)$$

Owing to the hexagonal geometry:

$$L = \sqrt{3}l \times m \qquad (8)$$

$$E_{local} = \frac{L}{md}E_{bulk} = \frac{\sqrt{3}l}{d}E_{bulk} \qquad (9)$$

The local stiffness can be also represented by the porosity as:

$$E_{local} = \frac{\sqrt{3}l}{d}E_{bulk} = \frac{1}{1-P}E_{bulk} \qquad (10)$$

For the gelatin scaffolds in our study, the porosity was consistently ~ 95%, resulting in 20-fold increase in the local stiffness compared with the bulk stiffness.

**Porosity measurement**. The porosity of the scaffold was characterized through acquiring the weight of swollen porous scaffold $W_1$ and the weight of dried porous scaffold $W_2$. For scaffold with large porosity, the density of scaffold can be identified to be similar to the density of water, thus the weight of swollen porous scaffold can represent the volume of total scaffold, whereas weight of dried scaffold can represent the volume of skeletons in scaffold. Thus, the porosity can be

calculated according to equation (11):

$$Porosity = \frac{W_1 - W_2}{W_1} \qquad (11)$$

**Isolation of mouse bone marrow-derived macrophage.** Mouse BMDMs were prepared from healthy male mice (8–12 weeks old C57BL/6) as described[34]. In brief, the whole bone marrow was derived and cultured in culture dish (Corning), whereas the cells were induced to macrophage phenotype using DMEM medium containing 10% FBS, 1% Penicillin–Streptomycin solution and 15% L929 conditioned medium with MCSF for 7 days at 37 °C and 5% $CO_2$. This process routinely yielded a macrophage population of > 95% purity as assessed by flow cytometry for CD11b (eBioscience, M1/70, 1:100) and F4/80 (abcam, CI:A3-1, 1:400). The additional M1 induction was achieved by stimulating macrophages with LPS (50 ng ml$^{-1}$) (L3129 Sigma) and the M2 induction was achieved by inducing BMDMs with IL-4/IL-13 (20 ng ml$^{-1}$) (PeproTech) for 3 days.

**Cell culturing in 3D porous scaffold.** Mouse bone marrow-derived macrophages were washed twice in PBS and then suspended into cell solution with PBS after 7 days of induction. After cell counting, the cells were centrifuged and re-suspended in DMEM medium with 10% FBS and 1% Penicillin/Streptomycin solution at a final concentration of $5 \times 10^6$ cells per milliliter. At the same time, the 3D scaffolds were placed in a 3.5 cm dish and sterilized under the ultraviolet radiation for 2 h. Next, the cell solution was added to the 3D scaffold so that cells can be absorbed into the 3D scaffolds. And ~ 100 microliter of cell solution was sufficient for a $10 \times 10 \times 1$ (mm)$^3$ scaffold. Scaffolds loaded with cells were then placed into 37° C environment with 5% $CO_2$ for 2 h to allow cells attachment inside the scaffold. Finally, 2 ml of DMEM was added to the dish and to support cell culture for 3 days.

HDFs (purchased from China National Infrastructure of Cell Line Resource, passages 5–9) were harvested using trypsin-EDTA (Wisent) and suspended in PBS. Cells were centrifuged and re-suspended in DMEM medium with 10% FBS and 1% Penicillin/Streptomycin solution at a final concentration of $2 \times 10^6$ cells per milliliter. Then, cell suspension was dropped into sterilized scaffolds followed by 2 h to allow cells attachment. Then, sufficient medium was supplied and cells were cultured up to 3 days. Experiments for cellular responses included six groups of scaffolds including two different pore sizes (i.e., 30 μm or 80 μm) and three different stiffness (i.e., 20 kPa, 70 kPa, or 190 kPa measured by AFM).

**RNA isolation and quantitative RT-PCR.** Total RNA extraction reagent (Vazyme) was used to isolate the RNA in each group. Moreover, for 3D cultured cells inside scaffolds, scaffolds were removed after total RNA extraction reagent treatment for 10 min. Polymerase chain reaction (PCR) with a Hiscript II qRT SuperMix Kit (V) (Vazyme) to reverse transcript 500 ng of RNA into cDNA. Levels of gene expression was measured with real-time PCR using AceQ qPCR SYBR green master mix (Vazyme), specific primers and CFX96 machine (Bio-Rad). Relative gene expression was quantified using the $2^{-\Delta\Delta Ct}$ methods and internally normalized to *Gapdh* expression as housekeeping gene, which was shown to be consistent with other housekeeping gene (i.e., Ubiquitin C) (Supplementary fig. 21b). For fibroblast activation, we quantified α-SMA (αSMA, ACTA2) and type I collagen (COLI); For different phenotypes of macrophages, we quantified pro-inflammatory factors like Interleukin-6 (Il-6), Interleukin-1β (Il-1β), tumor necrosis factor α (Tnfα), and anti-inflammatory factors like Interleukin-10 (Il-10).

The primer we used as followed:

| Primer name | Forward sequence | Reverse sequence |
|---|---|---|
| H-ACTA2 | GTGTTGCCCCTGAAGAGCAT | GCTGGGACATTGAAAGTCTCA |
| H-COL 1A1 | GAGGGCCAAGACGAAGACATC | CAGATCACGTCATCGCACAAC |
| H-GAPDH | GGAGCGAGATCCCTCCAAAAT | GGCTGTTGTCATACTTCTCATGG |
| m-Il-1β | GCAACTGTTCCTGAACTCAACT | ATCTTTTGGGGTCCGTCAACT |
| m-Il-6 | TAGTCCTTCCTACCCCAATTTCC | TTGGTCCTTAGCCACTCCTTC |
| m-Il-10 | CTTACTGACTGGCATGAGGATCA | GCAGCTCTAGGAGCATGTGG |
| m-Tnfα | CAGGCGGTGCCTATGTCTC | CGATCACCCCGAAGTTCAGTAG |
| m-Ubiquitin C | CCCAGTGTTACCACCAAGAAG | CCCCATCACACCCAAGAACA |
| m-Gapdh | TCACCACCATGGAGAAGGC | GCTAAGCAGTTGGTGGTGCA |

**Immunofluorescence staining.** For 2D-cultured cells, the cells were fixed with 4% paraformaldehyde for 15 min, and washed by PBS. Then, fibroblasts were permeabilized with PBS containing 0.5% Triton X-100 (Sigma) for 10 min and blocked with PBS containing 5% bovine serum albumin (BSA) (Amresco) for 1 h. Next, first antibodies were used for immunostaining overnight: rabbit monoclonal α-SMA antibody (Abcam, E184, 1:400), rat monoclonal CD11b antibody (eBioscience, M1/70, 1:100), mouse MHC- antibody with phycoerythrin (PE) labeled (eBioscience, M5/114.15.2, 1:200), mouse monoclonal iNOS antibody (abcam, EPR16635-ChiM-IgG2b, 1:200), rat monoclonal Arginase-1 (eBioscience, A1exF5, 1:200), rat monoclonal F4/80 antibody (abcam, CI:A3-1, 1:400), rat monoclonal Ly6G

antibody (Biolegend, 1A8, 1:100), mouse monoclonal COL-1 antibody (abcam, COL-1, 1:200). Cells were then washed and incubated with the corresponding secondary antibodies if necessary, including Alex 647 Goat anti-Rabbit IgG (Earthox, 1:400), Alex 488 Goat anti-Rat IgG (Earthox, 1:400). And then nuclear were stained by Hoechst 33342 (Beyotime, 1:2000) for 10 min. For cells inside the 3D scaffolds, the $10 \times 10 \times 1$ (mm)$^3$ scaffolds together with the cells inside were fixed with 4% paraformaldehyde for 30 min, then the scaffolds were cut to five independent pieces, with the size of $10 \times 2 \times 1$ (mm)$^3$ for each, and permeabilized with PBS containing 0.5% Triton X-100 for 20 min and the following staining procedure were the same as described above.

**Flow cytometry analysis.** For 2D-cultured cells, the cells were digested with trypsin-ethylenediaminetetraacetic acid (EDTA) until most of cells detached from the substrate. Suspended cells were collected then and treated with the following reagents: 4% paraformaldehyde for 15 min; PBS containing 0.5% Triton X-100 for 10 min (only for fibroblasts); PBS containing 5% BSA for 1 h; PBS containing 5% BSA and certain first antibodies like rabbit monoclonal αSMA (Abcam, E184, 1:400), rat MHC-II-PE antibody (eBioscience, M5/114.15.2, 1:200), mouse iNOS antibody (abcam, EPR16635-ChiM-IgG2b, 1:200), rat Arginase antibody (eBioscience, A1exF5, 1:200); PBS containing certain second antibodies like Alex 647 Goat anti-Rabbit IgG. Finally, over 10000 immune-stained cells were analyzed for fluorescence intensity using Flow cytometer LSRFortessa SORP (BD). For cells cultured in 3D scaffold, the whole gelatin scaffolds were digested with PBS containing 0.5% Collagenase I (Life Technologies) and 0.5% Collagenase IV (Life Technologies) in 37 °C for 30 min. The scaffolds would degrade allowing the collection of cells inside the scaffolds for following process.

**Cell elongation/area evaluation.** To evaluate the cell area and elongation in 2D and 3D scaffold, the cell membrane was stained by Cell plasma membrane staining kit (Abcam). For 2D-cultured cells, 10 μm in-depth images were acquired using confocal microscope, and the cell area could be calculated through analyzing the projection area of each cell in the images. The cell elongation was defined as the ratio of long and short axes. Three independent images from three independent groups were acquired for a single group. For cells inside the 3D scaffolds, the scaffolds were cut to five independent pieces before staining. Then, images were taken with 20–100 μm in-depth and made maximum projection, and the cell area and elongation were analyzed by same methods.

**Osmotic pressure study.** Different concentrations of PEG 400 were introduced in the culture medium to tune the osmotic pressure. Specifically, 0%, 2%, 4%, 8%, 12% of PEG 400 was introduced into the culture media, which was sterilized for cell culture. After 3 days of culture, cells were washed and fixed for following analysis.

**Mice wound healing model establishment.** Eight-week-old balb/c mice ( ~ 20 g) were used in the establishment of wound healing model. Specifically, mouse was anaesthetized with isoflurane, and a round PMMA mold with radius of 4 mm was introduced to mark a skin area with same size. This area is on the back of mouse, and each mouse has two symmetric skin areas on the left or the right side. And the skin tissue in the area was fully removed to create the wound. Then, the skin around the wound was sutured using a PMMA ring with an internal diameter of 8 mm and an external diameter of 10 mm to stabilize the wound area. The mold was fixed with the surrounding skin tissue through suturing, thus the wound would not experience wound contraction, which may influence the regeneration process. After that, scaffolds were cut into round shape with 8 mm diameter as well and swollen with normal saline before it was used to cover the wound area. At last, Tegaderm$^{TM}$ film (3 M) was used to cover the wound area to prevent water loss and infection. To evaluate the wound healing rate, Tegaderm$^{TM}$ film was removed every 2 days. After observation, a new Tegaderm$^{TM}$ film would be applied to cover the wound area.

**Section of scaffold/tissue sample.** Paraffin embedded scaffolds were collected after three days of cell culturing or four days of implantation, and 4% of paraformaldehyde was applied to fix in vitro scaffolds for 15 minutes while implanted scaffolds were fixed overnight in 10% formalin after being embedded in optimal cutting temperature (OCT) or paraffin. Then, scaffolds were embedded into paraffin and sectioned with a thickness of 8 μm to enable the following staining process. Frozen embedded scaffolds were prepared for immunostaining, while implanted scaffolds were collected and immediately embedded into OCT Compound (Leica) in − 80 °C. And 8 μm-thick slices were prepared for following immunostaining.

**Estimation of RGD motif density in the gelatin scaffolds.** RGD motif is known to mediate cell attachment on the gelatin scaffolds, which are based on hydrolyzed collagen I derived from cold fish skin[24]. Collagen formation begins with the translation of 3α chain and the formation of procollagen triple helix, for collagen I, procollagen is made of 2 α1 chains and 1 α2 chain[35]. It is known that α1 chain contains two RGD motifs among a total length of 1464 amino acids and α2 chain contains 3 RGD motifs among a total length of 1366 amino acids, which means

that there are seven RGD motifs within one procollagen molecule, which contains ~ 4000 amino acids with molecular weight (MW) of ~ 300kDa[36]. The RGD motifs were preserved in gelatin after hydrolysis to mediate cell adhesion[37]. Therefore, for a gelatin scaffold in the size of $10 \times 10 \times 1$ (mm)$^3$, which is made of 4 mg of gelatin: The total molecular number of RGD motif in the scaffold would be estimated as (where $N_A$ is defined as Avogadro constant):

$$N_{RGD} = \frac{Weight_{gelatin}}{MW_{procollagen}} \times 7 \times N_A = 3.6 \times 10^{16} \quad (12)$$

Considering the porosity of 95%, the volume for gelatin inside can be calculated:

$$V_{gelatin} = V_{scaffold} \times 5\% = 5 \times 10^{18} \, nm^3 \quad (13)$$

The average volume containing one single RGD motif would be estimated as:

$$V_{RGD} = \frac{V_{gelatin}}{N_{RGD}} = 140 \, nm^3 \quad (14)$$

The average spacing between nearby RGD motifs would be estimated as:

$$Gap_{RGD} = \sqrt[3]{V_{RGD}} \approx 5 \, nm \quad (15)$$

**Statistics**. Statistical analyses were performed using one-way or two-way analysis of variance (ANOVA) with the GraphPad Prism 7.0. Multiple comparison between the groups was performed using post hoc Tukey's multiple comparisons test (for one-way ANOVA) and Sidak's multiple comparisons test (for two-way ANOVA). For experiments comparing two groups, we performed a two-tailed paired Student's $t$ test. Sample sizes ($n$) are shown in captions and original data are shown in source data file. Differences were represented by $p$ value, which was available upon each analyzed data. Data were expressed as means ± s.d or means ± s.e.m.

**Reporting summary**. Further information on experimental design is available in the Nature Research Reporting Summary linked to this article.

## Data availability

The data in this work are available in the manuscript or Supplementary Information, or available from the corresponding author upon request. The source data for Figs. 1c, 2a, 2b, 2c, 2e, 3b, 3c, 3e, 3f, 4e, 4f, 4g, 4h, 4i, 4j, 5c, 5d, 5e, 5f, 5i, 5j, 5k, 5l, 6d, 6h, 6i, and Supplementary Figs. 1, 2b, 3b, 4a, 5a, 5b, 6a, 8a, 8b, 8d, 9a, 9b, 11c, 11d, 12a, 12b, 13d, 13e, 14c, 14d, 15, 16b, 18, 20a, 21b, 22b, 23 are provided as a Source Data file.

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

## Acknowledgements

We acknowledge professor Yan Shi and Dr. Tie Xia in School of Medicine, Tsinghua University for their kind assistance in AFM measurement as well as the Animal Core Facility and Center of Biomedical Analysis at Tsinghua University for technical assistance. This work is financially supported by National Key R&D Program of China (2017YFA0104901), National Natural Science Foundation of China (31671036), and Beijing Natural Science Foundation (JQ18022).

## Author contributions

S.J. and C.L. contribute equally to this work. S.J. and Y.D. conceived and designed the experiments; C.L. assisted in experiment design, BMDMs-related experiment and article writing; P.Z. assisted in fabrication of gelatin scaffolds and AFM test; W.L. and W.K. assisted in immunostaining and qPCR. C.H. guided the establishment of the mouse wound healing model; S.J. and Y.D. wrote the manuscript, which C.H. and G.G. helped to revise, Y.D. is the principal investigator of the supporting grants.

## Additional information

**Competing interests:** The authors declare no competing interests.

