## [Peer Review File · Nature Communications]

Reviewers' comments:

Reviewer #1 (Remarks to the Author):

This is very interesting research and of general interest for the material scientist working in the area of biomaterials. I have few comments on the work reported.

1. The compression test methods lacks some details, particularly size and shape of the probe the compression was performed with doesn't specified.
2. For the mechanical test the samples where produced in the different size and shape of molds thus the mechanical properties of the final sample will be different and could not be comparable, as freezing process is very complex and depends on many factors (sample volume and size, the mold material etc).
3. Porosity calculation is very simplified and actually represents the amount of water in the scaffold not porosity. This method doesn't take into account the water associated with polymer matrix (strong bound water, weak bound water and water in the swollen polymer walls). Authors need to have a look on the already established methods for the porosity characterisation in cryogels.
4. Pore size determination using SEM images is not straight forward process as has been shown previously by other researchers. SEM image represents a 3D view of the sample which is not easy to process with ImageJ see previous works by Savina and Gunko. A little bit more details on the analysis: image processing, what plugin have been used and how the pores were measured will be useful.
5. Authors are presenting the work on the cryogelation and effect of the cryoprotectant, however the introduction and the discussion insufficiently consider the previous studies (macroporous cryogels are known for at least 20 years). A lot of research papers and reviews are available on the cryogels preparation, analysis and control of the porosity. Only one has been cited in this work, so it is not clear if the authors has done proper analysis of the previous research before making any conclusions in this work. There are some work already published on the use of DMSO, methanol and other additives (by Galaev, Okay and others) which will be useful to consider and acknowledge if appropriate.
6. Videos 2 are not provided.
7. More details on Gelatine need to be provided (bovine, fish, modified or not) as it will have considerable effect on the final product.

Reviewer #2 (Remarks to the Author):

Jiang et al describe a strategy that uses cyroprotectants such as DMSO to control cryogel scaffold porosity independently of mechanics. The authors provide a detailed methodology and proposed mechanism in which DMSO is excluded during ice crystal formation leading to concentration and control over ice crystal growth. They perform extensive mechanical and structural analyses of the role of DMSO, and biocompatibility studies of fibroblast and macrophage activation in vitro and in a skin wounding model in vivo. They find that DMSO concentrations between 1-5% can control porosity without affecting bulk mechanics. In turn, mechanics can be adjusted by crosslinker concentration/time. Cell responses varied by porosity and stiffness, with smaller pores driving increased MHCII and IL1b expression by macrophages and less cell spreading in fibroblasts. Pore size appeared to exert the greatest differences. This study provides a useful and comprehensive method to control the cryogelation process and has many possible uses in tissue engineering. There are a few unanswered questions regarding scaffold characterization, and the biological response characterization could be expanded.

Results

1. Figure 1: Please specify what polymer and crosslinker was used to generate these images. (gelatin and GA?)

2. What was the effect of DMSO concentration on local modulus via AFM? This data is critical to interpret the cell mechanosensing data. From what I can tell only bulk modulus was characterized with differing DMSO concentrations.
3. As the authors stated, freezing an aqueous solution increases solute concentration between ice crystals as they grow. This is leveraged to increase DMSO concentration to control pore size, but wouldn't it also increase gelatin and crosslinker concentration thus affecting mechanics? This wouldn't invalidate the methods, but would require a consideration of the final effective concentration to fine tune the mechanics while achieving desired pore size.
4. The caption for SFig 5 is difficult to follow, and the results for SFig 5 and 6 were not presented in the results section. This applies to other SFIGs as well.
5. Freezing rate seems to be a final variable controlling pore size where more rapid freezing leads to smaller ice crystals. How can freezing rate be controlled during this process, for example, in larger scaffolds?
6. How was it determined that all residual GA was removed across material types? Free and fixed GA can be difficult to remove and will have a profound effect on macrophage behavior. Likewise, were all reagents certified or tested as endotoxin free?
7. Figure 4E: what is the y-axis in the dot plots? A scatter parameter?
8. Consider standardizing the color scheme across figures (80 um is red in Fig 3 but green in Fig 4)
9. MHCII is not specifically an M1 skewed marker, and M2 polarized cells may also highly express it (since authors defined pro-inflammatory as M1-like). It would be more accurate to describe them as activated rather than pro or anti inflammatory based on this marker. Inducing MHCII can be useful in numerous contexts, however, and may be discussed rather than within the M1/M2 framework.
10. For M1 and M2 designation, additional immunolabeling (IF or FACS) or biochemical assays (e.g. iNOS vs Arginase) activity would be needed. Co-stimulatory molecules such as CD80/86 would complement MHCII. PCR data is useful, but most of the changes presented are modest in the context of M1/M2 polarization in vitro, and there was a paucity of M2 markers for comparison (IL10 by macrophages is very contextual and is dominant in "M2c"). Were M1 and M2 stimulated controls considered (i.e. IL4 for M2 and LPS+IFN γ for M1) on these scaffolds? This would provide context regarding the magnitude of these effects and whether they provide additional immunoregulation in the presence of soluble factors.
11. Please briefly mention the type of wound healing model used in the results to orient the reader.
12. Some of the supplemental figures could be combined, such as the histology images.
13. The in vivo characterization would benefit from more detail regarding cell infiltrate and phenotype, possibly additional IF staining. The skin wound model may close too quickly to fully evaluated foreign body response characteristics/cell infiltration, and a subcutaneous model may be more useful for longterm effects.
14. SFig 19: was there an effect on wound closure – what were untreated controls like? Differences appear very small, and it's undefined what the statistical indicators are comparing.
15. How was viability determined? Some of the cells in high PEG concentrations for example do not look healthy. They state viability was high in all groups but do not show data or how it was determined.

Discussion

16. The confinement experiments are interesting, but are difficult to extrapolate to a porous scaffold. Even 30 um pores are fairly spacious to most cells.
17. Are there other studies that have used cryoprotectant as a component of a cryogel? Please discuss if so.
18. How does this work compare to previous ones that found an effect of pore size on macrophage activity such as: Sussman et al. "Porous Implants Modulate Healing and Induce Shifts in Local Macrophage Polarization in the Foreign Body Reaction"

Methods

19. Please specify immunostaining conditions for immunofluorescence and flow cytometry such as antibody clones and dilutions.
20. GAPDH is highly variable across macrophage phenotypes and is often not a suitable reference gene on its own. The stability of GAPDH across conditions should be verified and compared to other references.
21. What scaffold group was the PCR data normalized to?
22. ANOVA was used to determine statistical differences. What posthoc test was used to compare groups?

Reviewer #3 (Remarks to the Author):

Comments:

Authors have synthesized gelatin and gelatin-hyaluronic acid cryogels using variable concentration of DMSO as a cryoprotectant and glutaraldehyde as a crosslinker for controlling the pore size and stiffness in the polymeric scaffold. Further, they have performed a systematic mechanobiological investigation of 3D porous scaffolds by characterization of physical properties and evaluating cell-materials interaction using human fibroblast and macrophages. Authors have shown a detailed study on the effect of cryoprotectant (DMSO) enabled structural control of porous scaffold, which provide experimental confidence to the use of cryoprotectant. This information is an update in current information about the mechanobiology of cryogel. However, some points need to take care by authors before acceptance. Please find my comments in the annotated manuscript.

Reviewer 1:

1. The compression test methods lacks some details, particularly size and shape of the probe the compression was performed with doesn't specified.

Thanks for this comment. Briefly, the test was performed with a column-shape probe. The radius of its cross section is 20 mm, which is much larger than the samples. More detailed descriptions have been added to the methods as well.

2. For the mechanical test the samples where produced in the different size and shape of molds thus the mechanical properties of the final sample will be different and could not be comparable, as freezing process is very complex and depends on many factors (sample volume and size, the mold material etc.).

We thank the reviewer for this valuable comment. In order to eliminate the variance caused by the molds, all the samples in the main text were prepared in the same molds, which were cut to specific sizes for mechanical characterization.

3. Porosity calculation is very simplified and actually represents the amount of water in the scaffold not porosity. This method doesn't take into account the water associated with polymer matrix (strong bound water, weak bound water and water in the swollen polymer walls). Authors need to have a look on the already established methods for the porosity characterization in cryogels.

We thank the reviewer for this suggestion. In order to avoid the effect of water associated with polymer matrix on porosity measurement, we used an alkane reagent, namely cyclohexane, as a pore filling reagent, which exhibits very low affinity to gelatin. As shown blow, Gelatin is almost insoluble in cyclohexane after 30s vortex and incubation at room temperature for 30min. The porosity can be calculated by the following formula:
$$Porosity = \frac{V_{pores}}{V_{dry} + V_{pores}} = \frac{(W_{absorbed} - W_{dry}) / \rho_{solvent}}{W_{dry} / \rho_{gelatin} + (W_{absorbed} - W_{dry}) / \rho_{solvent}}$$

The density of cyclohexane ($\rho_{solvent}$) is 0.779 g/cm³, and the density of the dry scaffold approximately equals to the density of gelatin ($\rho_{gelatin}$, 1.037 g/cm³). The experimental results showed that the porosity did not exhibit significant difference among all the scaffolds, which is ~95% (93%-97%). The conclusion is consistent with the previous porosity estimation based on water.

4. Pore size determination using SEM images is not straight forward process as has been shown previously by other researchers. SEM image represents a 3D view of the sample which is not easy to process with

ImageJ see previous works by Savina and Gunko. A little bit more details on the analysis: image processing, what plugin have been used and how the pores were measured will be useful.

As suggested, detailed methods for pore size characterization have been added to the methods part. Basically, we measured the longest and shortest diameter of each pore in ImageJ and used the average as the final diameter of that pore, then 20 pores were selected randomly in each image while 9 images were used for each DMSO concentration to acquire the final result. Detailed methods for pore size characterization have been added to the methods part.

5. Authors are presenting the work on the cryogelation and effect of the cryoprotectant, however the introduction and the discussion insufficiently consider the previous studies (macroporous cryogels are known for at least 20 years). A lot of research papers and reviews are available on the cryogels preparation, analysis and control of the porosity. Only one has been cited in this work, so it is not clear if the authors has done proper analysis of the previous research before making any conclusions in this work. There are some work already published on the use of DMSO, methanol and other additives (by Galaev, Okay and others) which will be useful to consider and acknowledge if appropriate.

We thank the reviewer for this comment, we have added more references regarding previous works related to crogels as below, in particular cryogel production involving DMSO.

Several previous studies did supplement DMSO as a component in the pre-polymer solution prior to crosslinking for cryogel preparation. In contrast, DMSO was previously used as a solvent to dissolve other substances such as PNIPAAm¹, PLGA² and N-vinylcaprolactam³ for scaffold fabrication. Our study represents the first attempt of applying DMSO to control the pore size of the cryogel. Furthermore, the mechanism for DMSO-based pore-size control is firstly proposed and verified in our research.

6. Videos 2 are not provided.

Sorry, there was a mistake during the previous submission. We have provided the Videos 2 this time.

7. More details on Gelatine need to be provided (bovine, fish, modified or not) as it will have considerable effect on the final product.

The product information has been provided in the method now, the gelatin is from cold water fish skin(MW~60kDa), produced by sigma-Andrich, Cas. 9000-70-8

Reviewer 2:

1. Figure 1: Please specify what polymer and crosslinker was used to generate these images. (gelatin and GA?)

The results in Figure 1 were obtained by using gelatin scaffolds prepared with cold fish gelatin and GA as crosslinker.

2. What was the effect of DMSO concentration on local modulus via AFM? This data is critical to interpret

the cell mechanosensing data. From what I can tell only bulk modulus was characterized with differing DMSO concentrations.

We thank the reviewer for this valuable suggestion. To evaluate DMSO concentration on local modulus, we prepared the scaffolds with different DMSO concentrations (1%,3%,5%,7%,10%) using 0.1% GA for 16h crosslinking. The local modulus was detected by AFM. When the DMSO concentration is in the range of 1% to 5%, the local moduli of the scaffolds were all around 80 kPa, which did not show significant difference ($P=0.4069$). Interestingly, when the DMSO concentration was increased to 7% and more, the local modulus was reduced to less than 15 kPa ($P<0.001$). Since we used DMSO concentration in the range of 1-5% in this study to control the pore size, which did not significantly affect the both the bulk and local modulus of the porous scaffold, we achieved separate regulation of the stiffness and pore size.

3. As the authors stated, freezing an aqueous solution increases solute concentration between ice crystals as they grow. This is leveraged to increase DMSO concentration to control pore size, but wouldn't it also increase gelatin and crosslinker concentration thus affecting mechanics? This wouldn't invalidate the methods, but would require a consideration of the final effective concentration to fine tune the mechanics while achieving desired pore size.

We thank the reviewer for this valuable question. As the ice crystal form, the concentration of all components in the solution will be enriched including DMSO, glutaraldehyde, and gelatin. Meanwhile, introduction of DMSO in the cryogelation system will theoretically have inhibitory effect on the ice crystal formation, which will reduce the final concentration of glutaraldehyde and gelatin compared with the DMSO-free counterpart, resulting in decreased stiffness. In order to evaluate the effect of DMSO on scaffold stiffness, we performed AFM test as presented in the previous question. These data indicated that we could ignore the mechanical variance due to the slight changes in crosslinking in low DMSO concentration range (initial DMSO concentration between 1% to 5%). Meanwhile, if higher concentrations of DMSO were applied (e.g. initial DMSO concentration >7%), the DMSO would result in scaffold with lower stiffness. If needed, we could tune the initial concentration of GA to make up for the changes due to the high DMSO concentrations. We have included this concern in the discussion part.

4. The caption for SFig 5 is difficult to follow, and the results for SFig 5 and 6 were not presented in the results section. This applies to other SFIGs as well.

We have rewritten the relevant results and captions. We renumbered SFig5 to be SFig20 with caption as 'The cooling rate of the cryogelation process could be regulated by altering the size of the mold'. And we renumbered Sfig6 to be Sfig21 with caption as 'DMSO regulation on pore size was affected by reaction temperature'.

5. Freezing rate seems to be a final variable controlling pore size where more rapid freezing leads to smaller ice crystals. How can freezing rate be controlled during this process, for example, in larger scaffolds?

We thank the reviewer for this valuable suggestion. Freezing rate is the key regulator here especially for pore size control in larger scaffolds. Further improvement in freezing rate control can be achieved by using more efficient media for heat transfer (e.g. cryogelation in liquid freezing media instead of the current air cooling) and better tuning of crogelation temperature (e.g. test in lower temperatures or test gradient cooling).

6. How was it determined that all residual GA was removed across material types? Free and fixed GA can be difficult to remove and will have a profound effect on macrophage behavior. Likewise, were all reagents certified or tested as endotoxin free?

GA were removed through the introduction of sodium borohydride, which would react with residential GA to eliminate GA residue. Besides, we detected residual GA using HPLC modified from the protocol recommended by the United States Pharmacopeia. Specifically, HPLC operational parameters were listed as follows: Agilent 1200, DAD detector, InerSustain C18 column (5 μ m, 250 mm x 4.6 mm). H₂O: Acetonitrile = 35:65, flow rate: 1 ml/min, detection wavelength: 355 nm, column temperature: 40 ° C, injection volume: 20 μ L. Samples were extracted by ethanol with sonication. The standard curve was made by detecting a series of standard samples (0.01, 0.02, 0.05, 0.1, 0.5, 1, 10ppm) which showed that the detection limit is 0.02ppm. No characteristic peaks were detected in the samples made by different scaffolds, indicating that the content of glutaraldehyde is less than 0.02 ppm which is less than 0.0002% wt/wt normalized to sample weight.

To test the endotoxin content in the gelatin scaffolds, the endotoxin eluates from different scaffolds were prepared by incubation of the scaffolds in cell culture condition (37 °C, 5%CO₂) for 3 days and were detected by endotoxin kit (Genescript, L00350). Results showed that endotoxin eluates are all < 1EU/ml, which are just a little bit higher than FDA limitation for medical devices (\leq 0.5 EU/mL).

Meanwhile, the concentration of endontoxin(LPS) that has been reported to effectively induce macrophage M1 activation *in vitro* is around 25EU/ml (50ng/ml LPS , potency >500000EU/mg), which was also used in our study. Also, it has been reported that when endotoxin concentration is below 5EU/ml (10ng/ml LPS , potency >500000EU/mg), no significant induction of macrophage activation was observed *in vitro*⁴. Therefore, we expect that endotoxin content of materials(<1EU/ml) used in this study is

negligible for macrophage activation. Other materials used for cell culture were commercialized and were tested as endotoxin free.

7. Figure 4E: what is the y-axis in the dot plots? A scatter parameter?

It plots the cell size, which is represented by the FSC-A (Forward Scatter-Area) signal data from the flow cytometry.

8. Consider standardizing the color scheme across figures (80 µm is red in Fig 3 but green in Fig 4)

Thank you for your suggestion! We have standardized the color scheme across all figures.

9. MHCII is not specifically an M1 skewed marker, and M2 polarized cells may also highly express it (since authors defined pro-inflammatory as M1-like). It would be more accurate to describe them as activated rather than pro or anti inflammatory based on this marker. Inducing MHCII can be useful in numerous contexts, however, and may be discussed rather than within the M1/M2 framework.

We thank the reviewer for this great suggestion! We have revised our description on MHC-II expression. In order to further explore the M1/M2 phenotype regulated by scaffold properties, we have further incorporated another pair of M1/M2 markers — iNOS(M1)/Arginase(M2).

iNOS was used as a typical M1(pro-inflammatory) marker and more iNOS-positive cells could be observed in scaffolds with soft (i.e. 20 kPa and 70 kPa) and small (i.e. 30 µm) pores. In contrast, more Arginase1-positive cells indicating M2 (anti-inflammatory) phenotype could be observed in scaffolds with stiffer(i.e. 190 µm) and larger pores(i.e. 80 µm). Besides, flow cytometry results have further validated the above-mentioned conclusions. More details have been added in discussion part and additional results were discussed in question 10 below.

10. For M1 and M2 designation, additional immunolabeling (IF or FACS) or biochemical assays (e.g. iNOS vs Arginase) activity would be needed. Co-stimulatory molecules such as CD80/86 would complement MHCII. PCR data is useful, but most of the changes presented are modest in the context of M1/M2 polarization in vitro, and there was a paucity of M2 markers for comparison (IL10 by macrophages is very

contextual and is dominant in “M2c”) Were M1 and M2 stimulated controls considered (i.e. IL4 for M2 and LPS+IFN γ for M1) on these scaffolds? This would provide context regarding the magnitude of these effects and whether they provide additional immunoregulation in the presence of soluble factors.

We thank reviewer for great suggestions! As mentioned above, for M1/M2 designation another pair of marker — iNOS(M1)/Arginase(M2) were further incorporated in this study. Similar to MHC-II, iNOS was used as a M1 marker and more iNOS-positive cells could be observed in scaffolds with soft (i.e. 20 kPa and 70 kPa) and small (i.e. 30 μ m) pores. In contrast, more Arginase1-positive cells indicating M2(anti-inflammatory) phenotype could be observed in scaffolds with stiffer(i.e. 190 kPa) and larger pores(i.e. 80 μ m). Besides, flow cytometry results have corroborated this conclusion. More details and discussions have been added in manuscript.

To further demonstrate the significance in physical regulation of macrophage phenotypes, as suggested, the effect of additional immunoregulatory soluble factors were evaluated with 50ng/ml LPS for M1 induction and 20ng/ml IL-4 + 20ng/ml IL-13 for M2 induction. In 2D-culture control, elevation of iNOS (M1 marker) and Arginase-1(M2 marker) expression could be observed with M1 and M2 induction respectively.

When BMDM were cultured in scaffolds with M1 induction, overall increase in iNOS level and decrease in Arginase1 level were observed compared with no treatment condition. However, more iNOS-positive cells could be observed in scaffolds with smaller (i.e. 30 μ m) pores, though there was no significance difference when pore stiffness changed. Contrary to this, most Arginase1-positive cells could be observed within stiffer (i.e. 190 kPa) and larger pores (i.e. 80 μ m), whereas few Arginase1-positive cells could be observed within soft (i.e. 20 kPa) and small (i.e. 30 μ m) pores. Consistent with the immunofluorescent staining, flow cytometry results showed that there were highest number of Arginase1-positive cells (32.5%) and lowest

number of iNOS-positive cells (2.33%) in scaffolds within stiffer(i.e. 190 kPa) and larger pores(i.e. 80 μm) and nearly least Arginase1-positive cells(8.7%) within scaffolds with soft (i.e. 20 kPa) and small (i.e. 30 μm) pores.

When BMDM were cultured in scaffolds with M2 induction in medium, there was a sharp decrease in iNOS(M1) level and overall increase in Arginase1(M2) level. Still, cells in scaffolds with soft (i.e. 20 kPa) and small (i.e. 30 μm) pores showed the highest iNOS level with ~22% iNOS-positive cells and few iNOS-positive cells could be observed with stiffer and larger pores. Although cells in all group have high percent of Arginase1-positive cells(~70%), cells in scaffolds with soft (i.e. 20 kPa) and small (i.e. 30 μm) pores have lowest arginase1 level (~32%). Consistent with the immunofluorescent staining, flow cytometry results showed that there were few iNOS-positive cells (<0.2%) and more Arginase1-positive cells in scaffolds with larger pores(e.g. 66.9% in 20 kPa, 80 μm vs 43.9% in 20 kPa, 30 μm ; 57.9% in 190 kPa,80 μm vs 53.7% in 190 kPa, 30 μm). These results convincingly proved that the physical properties of the scaffolds, namely the pore stiffness and pore size, could regulate macrophage phenotype. Specifically, macrophages in scaffolds with small and soft pores tend to be activated towards M1(pro-inflammatory phenotype), whereas cells in scaffolds with larger and stiffer pores are prone to M2(anti-inflammatory phenotype) activation. The regulation on macrophages by the physical properties of the scaffolds is still significant even in the presence of additional immunoregulatory soluble factors.

Since we have systematically characterized the macrophage phenotype regulation by scaffolds physical properties through immunostaining and flow cytometry, we have put the current gene expression data into supplementary information.

11. Please briefly mention the type of wound healing model used in the results to orient the reader.

We have included more information in the manuscript. Briefly, a rounded wound with a radius of 5 mm was created on the back of Balb/c mice (8 weeks old). Then, scaffolds were shaped into the same size as the wound and applied to cover the wound area, following by the isolation of the wound through Tegaderm™ to prevent infection or water loss.

12. Some of the supplemental figures could be combined, such as the histology images.

Thank you for your suggestion! We have combined some supplemental figures.

13. The in vivo characterization would benefit from more detail regarding cell infiltrate and phenotype, possibly additional IF staining. The skin wound model may close too quickly to fully evaluated foreign body response characteristics/cell infiltration, and a subcutaneous model may be more useful for longterm effects.

We thank the reviewer for the valuable suggestion. As suggested, we reexamined the foreign body responses in a subcutaneous model. Scaffolds with different pore size and stiffness were implanted under the back skin of 8-week-old balb/c mice. Foreign body response was evaluated after 4 and 8 days after implantation, which were designed for evaluation of early and late foreign body response respectively. A great number of macrophages(F4/80) and neutrophils(Ly6G) were found to infiltrate into scaffolds with

small pores (i.e. 30 μm) on day 4, especially scaffolds with small and soft pores (i.e. 20 kPa, 30 μm). While, fewer macrophages and neutrophils could be observed in scaffolds with stiff pores, indicating a pro-inflammatory regulation by scaffolds with soft and small pores in early stage of foreign body response.

Meanwhile, activated fibroblasts (α -SMA) and collagen-I were stained to evaluate wounding healing in late stage of foreign body response. Infiltration of the activated fibroblasts showed no significant difference among different groups. However, collagen deposition, indicating fibrotic responses, could be observed on the edge on implanted scaffolds with stiff pores, especially in scaffolds with stiff and large pores (i.e. 190 kPa, 80 μm). These results indicated that scaffolds with small and soft pores (i.e. 20 kPa, 30 μm) could induce pro-inflammatory response shortly after implantation. In contrast, scaffolds with large and stiff pores (i.e. 190 kPa, 80 μm) would reduce inflammatory response in early stage but may induce fibrotic responses which facilitates the pro-wound healing effect.

14. SFig 19: was there an effect on wound closure – what were untreated controls like? Differences appear very small, and it's undefined what the statistical indicators are comparing.

We thank the reviewer for this valuable question. We designed this experiment to compare the effects of scaffolds' physical properties in regulation of cellular behaviors as well as the wound closure, so no untreated control was included, which is expected to eventually recover according to previous research⁶. We agree it will be more appropriate to compare the effects of scaffolds' physical properties in terms of the foreign body reaction (FBR) using a subcutaneous model as suggested the reviewer. We therefore further expanded the comparative investigation to the FBR using a subcutaneous model.

The statistical significance here represents the differences between the 20 kPa 80 μm group and the 190 kPa, 30 μm group. More details of the statistical indicators have been included in this figure.

15. How was viability determined? Some of the cells in high PEG concentrations for example do not look healthy. They state viability was high in all groups but do not show data or how it was determined.

After 3 days' culture, the viability of cells were evaluated by live/dead staining (Live cells: stained by Calcein-AM; green; dead cells: stained by PI; red). Few dead cells could be detected even PEG concentration was elevated up to 12%. Under the same imaging condition, the intensity of green fluorescence decreased a bit when PEG concentration is above 8%. Since the live cells were visualized by Calcein AM staining based on intracellular esterase activity in this kit, these results indicated that high PEG concentration might partly inhibit intracellular metabolic activities but has little effect on cell viability.

16. The confinement experiments are interesting, but are difficult to extrapolate to a porous scaffold. Even 30 μm pores are fairly spacious to most cells.

Cells can sense the porous structure of the ECM larger than their sizes. In our study, cell spreading could be inhibited by 30- μm pores that can induce different phenotypes (Fig. 3, Fig.4). This can be also supported by previous studies, for example, spatial confinement by 30- μm pores could significantly influence the early stage inflammatory response of macrophages *in vitro*⁵. In another study, macrophage phenotype could be dramatically varied in term of Foreign body reaction when the mean pore size of scaffold change from 160 μm to 34 μm ⁶. Moreover, scaffolds with smaller pore size (e.g. 20 μm in supplementary Fig.S4,S20,S21) can be made by controlling parameters, such as freezing rate.

17. Are there other studies that have used cryoprotectant as a component of a cryogel? Please discuss if so.

Several previous studies did use DMSO or other cryoprotectant as a component in the pre-crosslinking solution in cryogel preparation¹, which was used as a solvent to introduce other components into the scaffold. It is the first time that DMSO is applied just to control the pore size of the cryogel. Furthermore, the mechanism for DMSO-based controlled pore size is proposed and verified for the first time in our research.

18. How does this work compare to previous ones that found an effect of pore size on macrophage activity

such as: Sussman et al. “Porous Implants Modulate Healing and Induce Shifts in Local Macrophage Polarization in the Foreign Body Reaction”

Sussan et al. controlled the pore sizes in hydrogels (i.e. 34 μm and 160 μm) using particle templates. They only investigated the pore size effect on macrophage phenotypes *in vivo* and found that the greatest polarization (almost exclusively M1+M2⁻) was observed within 34 μm pores during foreign body reaction (FBR). The large pore in their study was much larger than the pore size range in our study and there is no systemic *in vitro* investigation in their work.

In our study, we realized separated control of structural and mechanical properties in porous scaffold. We examined the response of macrophages to both the pore size and stiffness *in vitro* and *in vivo*. In particular, the *in vitro* mechanism investigation in our study provided more insights on the regulatory schemes.

19. Please specify immunostaining conditions for immunofluorescence and flow cytometry such as antibody clones and dilutions.

More detailed information has been included now.

20. GAPDH is highly variable across macrophage phenotypes and is often not a suitable reference gene on its own. The stability of GAPDH across conditions should be verified and compared to other references.

Thank you for your suggestion! GAPDH is characterized again compared to other reference gene. In particular, Ubiquitin C can be a good housekeeping gene for mouse macrophages according to the previous study⁷.

When normalized to Ubiquitin C, GAPDH expression was verified to be similar for scaffolds of different pore size and stiffness Therefore, GAPDH and Ubiquitin C can both be used as suitable reference genes in our study.

21. What scaffold group was the PCR data normalized to?

Sample from 2D cultured macrophages was taken as the control group for normalization in all PCR experiments.

22. ANOVA was used to determine statistical differences. What posthoc test was used to compare groups?

Data from the groups were compared by one-way ANOVA or two-way ANOVA to determine statistical differences. Multiple comparison between the groups was performed using posthoc Tukey's multiple comparisons test (for one-way ANOVA) and Sidak's multiple comparisons test (for two-way ANOVA).

Reviewer 3:

1, It is highly recommended to revise the manuscript for English language/grammatical errors, typo-errors.

Thank you for your suggestion. The manuscript has been revised for language editing by Wiley Editing Service.

2, Should be supported by reference. One potential reference is; Modulated crosslinking of microporous polymeric cryogel affects in vitro cell adhesion and growth (2013), Macromolecular Biosciences, 13(7) 838-850.

Thank you for your suggestion. We have included this reference.

3, DMSO+ water based porous cryogels has already been reported in previous studies. Authors must elaborate the meaning of 'new strategy'. Please refer to the below articles published using similar solvent systems.

1.Synthesis and Characterization of a Temperature-Responsive Biocompatible Poly(N-vinylcaprolactam) Cryogel: a Step Towards Designing a Novel Cell Scaffold (<https://doi.org/10.1163/092050609X12457418891946>)

2.Okay O., Lozinsky V.I. (2014) Synthesis and Structure–Property Relationships of Cryogels. In: Okay O. (eds) Polymeric Cryogels. Advances in Polymer Science, vol 263. Springer, Cham pp 103-158.

3.X. Z. Zhang and C. C. Chu, Chem. Commun., 2003, 12, 1446

We thank the reviewer to list other related studies involving DMSO in cryogel fabrication. The function of DMSO in these studies is different from our study:

1), The first article used DMSO as a solvent for PVCl to fabricate PVCl-gelatin cryogel. No experiment was done to investigate the influence of DMSO to the cryogel.

2), The second article is a book chapter, where DMSO was mainly used as a solvent in most listed studies as well. There is one related research listed in the chapter conducted by M. Murat Ozmen, Oguz Okay, titled Formation of macroporous poly(acrylamide) hydrogels in DMSO/water mixture: Transition from cryogelation to phase separation copolymerization.

This work investigated the influence of DMSO to macroporous hydrogel structure. However, the only conclusion from the study is that high DMSO concentration (>25%) can cause dramatical structural change because of phase separation which basically ruined the typical porous structures as shown below. (Typical

structure with 60% DMSO, left SEM image). In contrast, we achieved the precise control of pore size (right SEM image) with a relatively low DMSO concentration (0%-5%), which has not been achieved in any study listed in this book chapter.

3), In the third article, anhydrous DMSO is applied as the solvent as well for the preparation of Thermosensitive PNIPAAm cryogel to dissolve the crosslinking precursor, which did not show structural regulatory effect of the cryogel.

Our study represents the first attempt of applying DMSO to control the pore size of the cryogel. Furthermore, the mechanism for DMSO-based pore-size control is firstly proposed and verified in our research. And some discussions comparing our study to previous work have been added in discussion part.

4, Authors used a word 'discovered' for the use and role of DMSO in cryogelation process, which is not justifying because there are several studies on cryogel synthesis using DMSO. Please refer for detail information in Okay O., Lozinsky V.I. (2014) Synthesis and Structure–Property Relationships of Cryogels. In: Okay O. (eds) Polymeric Cryogels. Advances in Polymer Science, vol 263. Springer, Cham

Thanks for the suggestions! In this statement, we wanted to highlight that it was the first time to use DMSO as cryoprotectant to control pore size in cryogelation process. As stated in the former question, all the previous studies used DMSO as a solvent but not as a structural modifier. Our work represents the first effort to introduce DMSO in cryogel fabrication as a structural regulator for the pore size of the cryogel without inducing mechanical changes. In addition, the mechanism for pore size controlling has been proposed and verified for the first time in our study. Relevant discussions comparing our study to previous work have been added in discussion part.

5, Should be 'cryogelation of polymer'

Thanks and it has been corrected.

6, Authors must explain that how they concluded the concentration of DMSO will be 40% in the unfrozen liquid microzone at -20 degree. It would be helpful for the readers. Considering the fact that, authors have used a experimental system containing 10% DMSO (initial concentration), wherein 90% water (solvent) will convert in to ice crystals at -20 degree. Then 10% DMSO along with polymer precursor will be 40%?

The freezing point curve of DMSO—water solution was obtained from previous study (Fig.S1). When the temperature drops to a certain temperature (e.g. -20 degree), the water molecules form ice crystals, and the DMSO is gradually enriched in the unfrozen phase. Due to DMSO enrichment, the freezing point of the unfrozen phase would drop and ultimately reach -20°C, at which point the DMSO concentration would rise to about 40% and further ice crystal growth would be halted. According to the curve, when the temperature drops to -20 degree, regardless of the initial concentration (e.g. 1% or 5% or 10%), the ultimate DMSO concentration after enrichment in the unfrozen phase will always be about 40%, the difference among different groups is the duration to reach the equilibrium state. We also quantified the changes in DMSO concentration throughout the entire ice growing and melting process and found the same trend of DMSO enrichment to around 40% during ice growing (Figure 1c).

7, Is it possible to calculate the change in color density in order to define the concentration of DMSO?

It is possible to estimate the DMSO concentration change by calculating the color change, but this image-based method can only provide qualitative estimation. Therefore, we have applied a more quantitative method to characterize the DMSO concentration in Figure 1C.

8, There are many literature studies that suggest the pore size of the cryogel can easily controlled by just changing the temperature, for example, at low temperature (like -40 degree) the freezing occurs faster than the comparatively higher sub-zero temperature (like -10 degree), thus quick freezing causes smaller ice porogen formation result in smaller pores. Therefore, how authors believe that addition of new element (like DMSO) will be better option, also what about the residual impurities (if polymer phase is having more than 40% DMSO during cryo-concentration)?

We thank the reviewer to raise this question. It is well known that the pore size of the cryogel can be regulated by the gelation temperature, however, the polymerization reaction can be dramatically influenced by the temperature too, which may result in different mechanical properties. What we achieved in this study with DMSO is the separate control of stiffness and pore size, which is unachievable by simply temperature control. In addition, the DMSO is actually removed in the washing process after the scaffold formation since they cannot react with gelatin nor GA, and they were soluble in water.

9, Legend on x and Y-axis of figure 1C is not clear. For example; Y axis shows value of DMSO concentration in %? What is the meaning of time during ice formation/melting per min (on x-axis)?

Thanks for your advice. More detailed information is now included in the caption to clarify these issues.

10, It must be '2G'

Sorry for this mistake. It has been corrected.

11, Provide the molecular weight of gelatin and hyaluronic acid.

Thank you for your advices. According to the product information provided by Sigma, the MW of cold water fish skin gelatin is ~60 kDa and MW of hyaluronic acid is 1500kDa-7500kDa. And this information has been added in the method part.

12, Ref 28 do not support the statement. Rather this, use the below ref. which is more suitable as it used 4% gelatin for cryogelation in presence of GA and discussing the elasticity of scaffolds. Ref. Kathuria et al., Synthesis and characterization of elastic and microporous chitosan-gelatin cryogels for tissue engineering. Acta Biomaterialia (2009), 5, 406-418.

Thank you for your suggestion, this article has been referenced to support our methods here.

13, Provide a reference as a modified procedure from; Chang et al., Preparation and characterization of gelatin/hyaluronic acid cryogels for....., Acta Biomater, 2013, 9(11)9012-9026.

Thank you for your suggestion, the suggested reference is added now.

14, I believe that it should be 'Gelatin'!

Thanks, it has been corrected.

15, Please provide the pH of the solution.

The pH of 1% sodium borohydride solution is 9.7, and this information has been added into the methods.

16, It should be expressed either as $1 \times 1 \times 1$ (mm)³ or 1 mm³

Thank you for your suggestion, all the size expression have been modified.

17, Provide the magnification scale and spot size.

All the images for pore size determination is under 1000X. This information has been added to the methods.

18, Provide a reference or briefly mention the parameters used for plasma treatment.

Thanks for the suggestions. Parameters used for plasma treatment have been included now.

19, Is this appropriate word?

It should be debris.

20, Please replace with 'harvested'.

Thank you for your advice. It has been changed as suggested.

21, It is suggested to give an average weight of mice used in the study.

The average weight is 20g. This information has been included in the method now.

22, Please clear, whether paraffin embedding was done before formaldehyde fixation or vice-versa?

Paraffin embedding was done after formaldehyde fixation. This information has been included in the method now.

23, Abbreviation used in equation should be named and defined properly, like NA.

We thank the reviewer for this suggestion. All the abbreviation used in equation have been named and defined. For example, NA is defined as Avogadro constant.

References:

1. Zhang, X.-Z. & Chu, C.-C. Thermosensitive PNIPAAm cryogel with superfast and stable oscillatory properties. *Chemical Communications*, 1446-1447, doi:10.1039/B301423A (2003).
2. Ho, M.-H. et al. Preparation of porous scaffolds by using freeze-extraction and freeze-gelation methods. *Biomaterials* 25, 129-138, doi:https://doi.org/10.1016/S0142-9612(03)00483-6 (2004).
3. Srivastava, A. & Kumar, A. J. J. o. M. S. M. i. M. Thermoresponsive poly(N-vinylcaprolactam) cryogels: synthesis and its biophysical evaluation for tissue engineering applications. 21, 2937-2945, doi:10.1007/s10856-010-4124-3 (2010).
4. Rosenstreich D L , Vogel S N , Jacques A R , et al. Macrophage sensitivity to endotoxin: genetic control by a single codominant gene.[J]. *Journal of Immunology*, 1978, 121(5):1664.
5. Jain N, Vogel V. Spatial confinement downsizes the inflammatory response of macrophages[J]. *Nature Materials*, 2018, 17(12): 1134-1144.
6. Dunn, L. et al. Murine model of wound healing. *Journal of visualized experiments : JoVE*, e50265-e50265, doi:10.3791/50265 (2013).
7. Ziegler, C. et al. Identification of stable reference genes for lipopolysaccharide-stimulated macrophage gene expression studies. *Biology Methods and Protocols* 1, doi:10.1093/biomethods/bpw005 %. *Biology Methods and Protocols* (2016).

Reviewers' comments:

Reviewer #1 (Remarks to the Author):

Thanks for answer to comment 2. "For the mechanical test the samples were produced in the different size and shape of molds thus the mechanical properties of the final sample will be different and could not be comparable, as freezing process is very complex and depends on many factors (sample volume and size, the mold material etc.). "

"We thank the reviewer for this valuable comment. In order to eliminate the variance caused by the molds, all the samples in the main text were prepared in the same molds, which were cut to specific sizes for mechanical characterization."

Thanks for the answer. I appreciate that the approach used is good for the comparing the samples (made at the same molds) and finding the effect of other conditions. However I wanted to point that authors have to be careful to translate this directly to the samples made in different molds and used for other studies, as the compression modulus will be not the same.

Reviewer #2 (Remarks to the Author):

The authors have answered the majority of the reviewer's questions, though some final changes will need to be addressed.

2.1 – thank you for clarifying that GA and gelatin was used, but please also include this information in the caption for Figure 1 and also stated more explicitly in the results section (rather than e.g.)

2.4 – Sfig 4b. DMSO is labeled at the top of the panel (these are with other solvents, correct?)

2.5 – I think the importance of freezing rate can be included in a brief paragraph of the limitations of this methodology. This control is somewhat empirical and will strongly depend on other manufacturing conditions that affect the cooling rate (and potential solutions can be mentioned as well).

2.6 – Thank you for these analyses. If the GA concentration was below detection, I think it's reasonable to say that it was not detected at all (instead reporting the ppm and % as the limit of detection).

Likewise, please report the endotoxin results as well (preferably as EU/wt of scaffold). I would also caution conclusions of how much endotoxin is needed to 'activate' macrophages. Activation is a very subjective term and depends on experimental objectives. Endotoxin can be significant at very low concentrations such as 2 EU/ml (for example TLR mediated signaling). Even if a phenotype is not detected, macrophages can be primed endotoxin to potentiate additional stimuli. The endotoxin appears low, so simply reporting these values, such as in the supplement, would be sufficient.

2.7 – The axis should be labeled with FSC-A then (like iNOS is in the other panel). Other forward scatter parameters (e.g. FSC-W) can be used to indicate cell size as well. Also, please define the ratios used for MHCII etc. Is this essentially % of cells, and is it calculated from flow cytometry or imaging data? (indicate in caption).

Also, there are a lot events on the lower x-axis in the M2 staining conditions. Another look at the compensation or using a biexponential display may provide some insights.

2.13 – The expanded in vivo analysis is very important, but some details are missing that make it difficult to interpret. For example, what is the time point in Figure 6? Also, it's unlikely that a GA crosslinked material has degraded in only 8 days as suggested in SFig 17. Then, Sfig 19 shows a

largely intact scaffold histologically. Since the material was not fixated to the tissue, is it more likely that the material is falling off during the healing process? (Also, SFig17a, the 30um histology looks very different from all other 30um implant sections presented.) The major conclusion for in vivo response seems to be that the material does not interfere with healing, or induce a strong pro-inflammatory reaction.

And an additional question:

1 – During scaffold processing, cyrogels are prepared, washed, then frozen/lyophilized. Can the authors comment on the effect of this refreezing process where presumably large ice crystals form in 0% DMSO conditions? Are all SEM images after this refreeze and lyophilization process?

Reviewer #3 (Remarks to the Author):

Authors have addressed the comments carefully in the revised manuscript. It may be considered for publication after complying the journal guidelines.

Reviewer1:

I appreciate that the approach used is good for the comparing the samples (made at the same molds) and finding the effect of other conditions. However, I wanted to point that authors have to be careful to translate this directly to the samples made in different molds and used for other studies, as the compression modulus will be not the same.

Thank you for the valuable suggestions! We have noticed that samples made in different molds have different pore structure. We will pay more attention to this effect on mechanical properties in future studies!

Reviewer2:

2.1- thank you for clarifying that GA and gelatin was used, but please also include this information in the caption for Figure 1 and also stated more explicitly in the results section (rather than e.g.)

Thank you for the comments! We have included this information in the figure caption and the results section.

2.4- Sfig 4b. DMSO is labeled at the top of the panel (these are with other solvents, correct?)

Sorry for the mistake, these scaffolds were made with Glycerol or Methanol in Sfig 4b, not DMSO. We have corrected the labeling.

2.5- I think the importance of freezing rate can be included in a brief paragraph of the limitations of this methodology. This control is somewhat empirical and will strongly depend on other manufacturing conditions that affect the cooling rate (and potential solutions can be mentioned as well)

Thank you for the advice! We have discussed the limitations in discussion part.

“It should be noted that current approaches for cooling rate control are still limited due to the inefficient heat transfer by air cooling. Further improvement can be achieved by using more efficient media for heat transfer (e.g. cryogelation in liquid freezing media instead of the current air cooling) and temperature controlling instrument for gradient cooling.”

2.6-Thank you for these analyses. If the GA concentration was below detection, I think it's reasonable to say that it was not detected at all (instead reporting the ppm and % as the limit of detection).

Likewise, please report the endotoxin results as well (preferably as EU/wt of scaffold). I would also caution conclusions of how much endotoxin is needed to 'activate' macrophages. Activation is a very subjective term and depends on experimental objectives. Endotoxin can be significant at very low concentrations such as 2 EU/ml (for example TLR mediated signaling). Even if a phenotype is not detected, macrophages can be primed endotoxin to potentiate additional stimuli. The endotoxin appears low, so simply reporting these values, such as in the supplement, would be sufficient.

Thank you for your valuable comments. We have revised the description of GA and endotoxin residue detection in discussion part respectively, which are listed as below.

“In addition, GA residue was not detectable using HPLC modified from the protocol recommended by the United States Pharmacopeia, and endotoxin eluates from scaffolds showed low level (Supplementary figure 23)”. The results of endotoxin detection have been added in supplementary as

Sfig 23. In particular, endotoxin eluates from scaffolds were shown in Sfig 23b and endotoxin residue normalized to scaffolds' weight were shown in Sfig 23c, indicating that endotoxin on scaffolds ranged from 0.1 to 0.2EU/mg.

Response Figure 2.6 (Supplementary Figure 22), Endotoxin eluates from scaffolds. (a) Standard curve used for endotoxin detection. (b) endotoxin eluates from different scaffolds were prepared by incubation of the scaffolds in cell culture condition (37°C, 5%CO₂) for 3 days and were detected by endotoxin kit (Genescript, L00350). (c) Endotoxin residue normalized to scaffolds weight. Data are means ± s.e.m.

2.7- The axis should be labeled with FSC-A then (like iNOS is in the other panel). Other forward scatter parameters (e.g. FSC-W) can be used to indicate cell size as well. Also, please define the ratios used for MHCII etc. Is this essentially % of cells, and is it calculated from flow cytometry or imaging data? (indicate in caption).

Also, there are a lot events on the lower x-axis in the M2 staining conditions. Another look at the compensation or using a biexponential display may provide some insights.

Thank you for your suggestion! we have relabeled this FACS data. Ratio in Figure 4g, 4h, 4i and 4j are calculated from imaging data. This information has also been added to the figure caption.

Original flow cytometry results using biexponential display without gating have been added in **source data file**. The cluster of dots accumulated near zero point of axis may be caused by cell debris, which existed in all groups including control group without staining (as shown in Response Fig.2.7a below). These cell debris could be excluded from analysis by tuning SSC-A/FSC-A gates before iNOS / Arginase-1

gating. And we believe these dots did not influence the consistency of iNOS / Arginase-1 gating. (Fig.b, original data of Figure 4f)

Response Figure 2.7, Original flow cytometry results using biexponential display without gating. (a) Control group without antibody staining. (b) Original flow cytometry data of Figure 4f. These results have been added in source data file.

2.13- The expanded in vivo analysis is very important, but some details are missing that make it difficult to interpret. For example, what is the time point in Figure 6? Also, it's unlikely that a GA crosslinked material has degraded in only 8 days as suggested in SFigure 17. Then, Sfig 19 shows a largely intact scaffold histologically. Since the material was not fixated to the tissue, is it more likely that the material is falling off during the healing process? (Also, SFigure17a, the 30um histology looks very different from all other 30um implant sections presented.) The major conclusion for in vivo response seems to be that the material does not interfere with healing, or induce a strong pro-inflammatory reaction.

Thanks for the valuable comments. Figure 6c is based on scaffold implanted for 4 days as well as Figure 6f. The information has been added to the figure caption.

Meanwhile, we believed the scaffolds were partially degraded during 8 days of experiment in the wound healing model (in SFigure 17 and SFigure 18). The slowed scaffold degradation in Sfigure 19 is expected due to different skin model where the fibrous capsule may help retain the scaffolds' structure in the subcutaneous implantation model. In contrast, fast degradation due to the inflammation or the shrinkage of scaffolds due to wound size reduction might occur in the wound healing model. Furthermore, as for wound healing model, the scaffold should not fall because all the wound area is covered with Tegaderm film. During the whole process, the scaffold was observed every two days, to make sure that it is still there covering the wound.

We showed SFigure-17a aiming to illustrate few cells infiltrated into scaffolds after 2 days' post implantation which could be clearly observed with few cellular nuclear staining. The high intensity of red staining in SFigure-17a compared to other figures might be caused by variance in histological slice thickness, where thicker slice containing more gelatin scaffolds tended to be more extensively stained by eosin to show deeper red color, especially for scaffolds with small pores. We will pay attention to keep the thickness of H&E slice constant in the future.

We revised our conclusion for *in vivo* part. “Our results indicated that by adjusting the physical properties of the implanted porous 3D scaffolds, we could regulate *in vivo* cellular response *both in a* skin wound healing model and a subcutaneous model. Scaffolds with small and soft pores will induce high level of inflammatory response and scaffolds with large and stiff pores will promote fibrotic response and wound healing. Although there was no significant difference in wound healing rate among different groups, tuning the scaffolds’ physical properties could potentially be leveraged to modulate foreign body response and tissue regeneration.”

An additional question- During scaffold processing, cryogels are prepared, washed, then frozen/lyophilized. Can the authors comment on the effect of this refreezing process where presumably large ice crystals form in 0% DMSO conditions? Are all SEM images after this refreeze and lyophilization process?

All the SEM images are taken after the refreezing and lyophilization. Since the pores within scaffolds have been already fully formed before refreezing, the ice crystal formation during refreezing is expected to be confined by scaffolds’ porous structure, thus the refreeze and lyophilization is believed not to influence the pore structure. To verify this, we took confocal fluorescent images of the scaffolds (all with 60- μm projection) before and after refreezing and lyophilization, which indicated that the pore structure remained unchanged after the refreezing and lyophilization for all the scaffolds. These results have been discussed in discussion part and added in supplementary as SFigure 21.

Response Figure for additional question, Pore structure was not affected by washing, refreezing and lyophilization process. (a) confocal fluorescent images of the scaffolds before and after refreezing and lyophilization. The images were taken with 30 frames with z steps of 2 μm and made maximum projection. Scale bar: 50 μm (b) Pore sizes were analyzed by image J. Data are means \pm s.e.m. ($n > 30$, $p > 0.05$, two-tailed Student’s t-test)

Reviewer3:

Authors have addressed the comments carefully in the revised manuscript. It may be considered for publication after complying the journal guidelines.

Thank you for your comments and we appreciate your effort in helping us with editing our manuscript!

REVIEWERS' COMMENTS:

Reviewer #2 (Remarks to the Author):

The authors have done a wonderful job addressing these comments. I agree with the authors that contraction of the material during healing would explain the absence histologically (as glutaraldehyde crosslinked materials are typically very robust).

REVIEWERS' COMMENTS:

Reviewer #2 (Remarks to the Author):

The authors have done a wonderful job addressing these comments. I agree with the authors that contraction of the material during healing would explain the absence histologically (as glutaraldehyde crosslinked materials are typically very robust).

Thank you for your comments and we appreciate your effort in helping us with editing our manuscript!